# The sterol-responsive RNF145 E3 ubiquitin ligase mediates the degradation of HMG-CoA reductase together with gp78 and Hrd1

Sam A Menzies[†], Norbert Volkmar[†], Dick JH van den Boomen, Richard T Timms[‡], Anna S Dickson, James A Nathan, Paul J Lehner*

Department of Medicine, Cambridge Institute for Medical Research, Cambridge, United Kingdom

*For correspondence:
pjl30@cam.ac.uk

[†]These authors contributed equally to this work

Present address: [‡]Department of Medicine, Brigham and Women's Hospital, Boston, United States

Competing interests: The authors declare that no competing interests exist.

**Abstract** Mammalian HMG-CoA reductase (HMGCR), the rate-limiting enzyme of the cholesterol biosynthetic pathway and the therapeutic target of statins, is post-transcriptionally regulated by sterol-accelerated degradation. Under cholesterol-replete conditions, HMGCR is ubiquitinated and degraded, but the identity of the E3 ubiquitin ligase(s) responsible for mammalian HMGCR turnover remains controversial. Using systematic, unbiased CRISPR/Cas9 genome-wide screens with a sterol-sensitive endogenous HMGCR reporter, we comprehensively map the E3 ligase landscape required for sterol-accelerated HMGCR degradation. We find that RNF145 and gp78 independently co-ordinate HMGCR ubiquitination and degradation. RNF145, a sterol-responsive ER-resident E3 ligase, is unstable but accumulates following sterol depletion. Sterol addition triggers RNF145 recruitment to HMGCR via Insigs, promoting HMGCR ubiquitination and proteasome-mediated degradation. In the absence of both RNF145 and gp78, Hrd1, a third UBE2G2-dependent E3 ligase, partially regulates HMGCR activity. Our findings reveal a critical role for the sterol-responsive RNF145 in HMGCR regulation and elucidate the complexity of sterol-accelerated HMGCR degradation.

**Editorial note:** This article has been through an editorial process in which the authors decide how to respond to the issues raised during peer review. The Reviewing Editor's assessment is that all the issues have been addressed (see decision letter).

DOI: https://doi.org/10.7554/eLife.40009.001

## Introduction

Cholesterol plays a critical role in cellular homeostasis. As an abundant lipid in the eukaryotic plasma membrane, it modulates vital processes including membrane fluidity and permeability (*Hannich et al., 2011*; *Haines, 2001*) and serves as a precursor for important metabolites including steroid hormones and bile acids (*Payne and Hales, 2004*; *Chiang, 2013*). The cholesterol biosynthetic pathway in mammalian cells also provides intermediates for essential non-steroid isoprenoids and therefore requires strict regulation (*Goldstein and Brown, 1990*). The endoplasmic-reticulum (ER) resident, polytopic membrane glycoprotein 3-hydroxy-3-methylglutaryl coenzyme A reductase (HMGCR) is central to this pathway, catalysing the formation of mevalonate, a crucial isoprenoid precursor. As the rate-limiting enzyme in mevalonate metabolism, HMGCR levels need to be tightly regulated, as dictated by intermediates and products of the mevalonate pathway (*Johnson and DeBose-Boyd, 2018*). The statin family of drugs, which acts as competitive inhibitors of HMGCR, represents the single most successful approach to reducing plasma cholesterol levels and therefore preventing atherosclerosis-related diseases (*Heart Protection Study Collaborative Group, 2002*).

**eLife digest** Cholesterol is a fatty molecule that is essential for our health; for example, it is a component of the outer membrane that surrounds every cell in our body. Yet, it also has a reputation for clogging arteries and causing heart attacks and strokes. Our organism can adjust the amount of cholesterol it creates through an enzyme called HMGCR, which is found in all cells. Switching off HMGCR, for instance by taking drugs called statins, reduces the amount of cholesterol made by cells. To regulate the activity of HMGCR, the body uses proteins known as E3 ubiquitin ligases, which can label the enzyme for destruction. However, the identity of the ligases that target HMGCR is a matter of intense debate.

Here, Menzies, Volkmar et al. addressed this issue by using an approach called a genome-wide CRISPR forward genetic screen. First, HMGCR was marked inside the cells with a fluorescent tag to watch how its levels change in response to different amounts of cholesterol. Then, each gene in the cell was deleted, and the effects recorded. This allowed Menzies, Volkmar et al. to find the genes responsible for the rapid destruction of HMGCR.

The experiments revealed that the E3 ubiquitin ligases RNF145 and gp78 are independently responsible for the degradation of the majority of HMGCR, with a third ligase, Hrd1, getting involved if the first two are absent. In particular, RNF145 builds up when a cell is starved of cholesterol, but it immediately marks HMGCR for destruction once cholesterol becomes more abundant. This ligase can therefore both sense and respond to the amount of cholesterol in a cell, making it a perfect candidate for regulating HMGCR based on what the body needs.

Identifying the proteins that adjust the levels of HMGCR sheds light on how a cell controls the amount of cholesterol it creates. This knowledge could be relevant in the fight against the health problems associated with this molecule.

DOI: https://doi.org/10.7554/eLife.40009.002

Understanding how HMGCR is regulated is therefore of fundamental biological and clinical importance.

Cholesterol, together with its biosynthetic intermediates and isoprenoid derivatives, regulates HMGCR expression at both the transcriptional and posttranscriptional level (*Johnson and DeBose-Boyd, 2018*). Low cholesterol induces transcriptional activation of HMGCR through the sterol response element binding proteins (SREBPs) which bind SREs in the promoter region (*Osborne, 1991*). In a cholesterol-rich environment, SREBPs are inactive and held in the ER in complex with their cognate chaperone SREBP cleavage-activating protein (SCAP) in association with the ER-resident Insulin-induced genes 1/2 (Insig-1/2) anchor proteins (*Dong and Tang, 2010*; *Yabe et al., 2002*). A decrease in membrane cholesterol triggers dissociation of the SCAP-SREBP complex from Insigs and translocation to the Golgi apparatus, where the SREBP transcription factor is proteolytically activated by Site-1 and Site-2 proteases, released into the cytosol and trafficked to the nucleus (reviewed in *Horton et al., 2002*). Low sterol levels therefore dramatically increase both HMGCR mRNA and extend HMGCR protein half-life, ensuring the resultant elevated enzyme levels stimulate the supply of mevalonate to re-balance cholesterol homeostasis (*Goldstein and Brown, 1990*; *Brown et al., 1973*). Once cholesterol levels are restored, excess HMGCR is rapidly degraded by the ubiquitin proteasome system (UPS) in a process termed sterol-accelerated degradation (*Hampton et al., 1996*; *Ravid et al., 2000*; *Sever et al., 2003a*). This joint transcriptional and translational regulation of HMGCR is controlled by a host of ER-resident polytopic membrane proteins and represents a finely balanced homeostatic mechanism to rapidly regulate this critical enzyme in response to alterations in intracellular cholesterol. While the ubiquitin-mediated, post-translational regulation of HMGCR is well-established, the identity of the critical mammalian ER-associated degradation (ERAD) E3 ubiquitin ligase(s) responsible for sterol-accelerated HMGCR ERAD remains controversial.

In yeast, *S. cerevisiae* encodes three ERAD E3 ubiquitin ligases, of which Hrd1p (HMG-CoA degradation 1), is named for its ability to degrade yeast HMGCR (Hmg2p) in response to non-sterol isoprenoids (*Hampton et al., 1996*; *Bays et al., 2001*). The marked expansion and diversification of E3 ligases in mammals makes the situation more complex, as in human cells there are 37 putative E3

ligases involved in ERAD, few of which are well-characterised (*Kaneko et al., 2016*). Hrd1 and gp78 represent the two mammalian orthologues of yeast Hrd1p. Hrd1 was not found to regulate HMGCR (*Song et al., 2005*; *Nadav et al., 2003*). However, gp78 was reported to be responsible for the sterol-induced degradation of HMGCR as (i) gp78 associates with Insig-1 in a sterol-independent manner, (ii) Insig-1 mediates a sterol-dependent interaction between HMGCR and gp78, (iii) overexpression of the transmembrane domains of gp78 exerted a dominant-negative effect and inhibited HMGCR degradation, and (iv), siRNA-mediated depletion of gp78 resulted in decreased sterol-induced ubiquitination and degradation of HMGCR (*Song et al., 2005*). The same laboratory subsequently suggested that the sterol-induced degradation of HMGCR was mediated by two ERAD E3 ubiquitin ligases, with TRC8 involved in addition to gp78 (*Jo et al., 2011*). However, these findings remain controversial as, despite confirming a role for gp78 in the regulation of Insig-1 (*Lee et al., 2006*; *Tsai et al., 2012*), an independent study found no evidence for either gp78 or TRC8 in the sterol-induced degradation of HMGCR (*Tsai et al., 2012*). Therefore, the E3 ligase(s) responsible for the sterol-accelerated degradation of HMGCR remain disputed.

The introduction of systematic forward genetic screening approaches to mammalian systems (*Carette et al., 2009*; *Wang et al., 2014*) has made the unbiased identification of E3 ubiquitin ligases more tractable, as demonstrated for the viral (*van den Boomen and Lehner, 2015*; *van de Weijer et al., 2014*; *Stagg et al., 2009*) and endogenous regulation of MHC-I (*Burr et al., 2011*; *Cano et al., 2012*).

To identify the E3 ligases governing HMGCR ERAD, we applied a genome-wide forward genetic screen to a dynamic, cholesterol-sensitive reporter cell line, engineered to express a fluorescent protein fused to endogenous HMGCR. This approach identified cellular genes required for sterol-induced HMGCR degradation, including UBE2G2 and the RNF145 ERAD E3 ubiquitin ligase. The subtle phenotype observed upon RNF145 depletion alone suggested redundant ligase usage. A subsequent, targeted ubiquitome CRISPR/Cas9 screen in RNF145-knockout cells showed RNF145 to be functionally redundant with gp78, the E3 ligase originally implicated in HMGCR degradation. We confirmed that loss of gp78 alone showed no phenotype, while loss of both E3 ligases significantly inhibited the sterol-induced ubiquitination and degradation of HMGCR. Complete stabilisation required additional depletion of a third ligase - Hrd1. We find that endogenous RNF145 is an auto-regulated, sterol-responsive E3 ligase which is recruited to Insig proteins under sterol-replete conditions, thus promoting the regulated ubiquitination and sterol-accelerated degradation of HMGCR. Our data resolve the controversy of the E3 ligases responsible for the post-translational regulation of HMGCR and emphasise the complexity of the mammalian ubiquitin system in fine-tuning sterol-induced HMGCR turnover and cholesterol homeostasis.

## Results

### Targeted knock-in at the endogenous HMGCR locus creates a dynamic, cholesterol-sensitive reporter

To identify genes involved in the post-translational regulation of HMGCR, we engineered a cell line in which Clover, a bright fluorescent protein (*Lam et al., 2012*), was fused to the C-terminus of endogenous HMGCR, generating an HMGCR-Clover fusion protein (*Figure 1A*). The resulting HMGCR-Clover Hela single-cell clone expresses a dynamic, cholesterol-sensitive fluorescent reporter that is highly responsive to fluctuations in intracellular cholesterol. Basal HMGCR-Clover levels in sterol-replete tissue culture media were undetectable by flow cytometry (*Figure 1B*) and phenocopy endogenous WT HMGCR expression (*Figure 1C*, compare lanes 1 and 4). Following overnight sterol depletion, a ~ 25 fold increase in HMGCR-Clover expression was detected (shaded grey to blue histogram in *Figure 1D*, *Figure 1C* (lanes 2 and 5)), representing a combination of increased SREBP-induced transcription and decreased sterol-induced HMGCR degradation. Reintroduction of sterols induced the rapid degradation of HMGCR-Clover (~80% decrease within 2 hr), confirming the sterol-dependent regulation of the reporter (blue to red histogram in *Figure 1D*). Residual, untagged HMGCR detected by immunoblot analysis in the reporter cells under sterol-depleted conditions suggested that at least one HMGCR allele remained untagged (*Figure 1C*, compare lanes 2 and 5), which was confirmed by PCR-amplification and sequencing of the genomic locus (*Figure 1—figure supplement 1A–C*). The unmodified allele allowed us to monitor both tagged and untagged forms

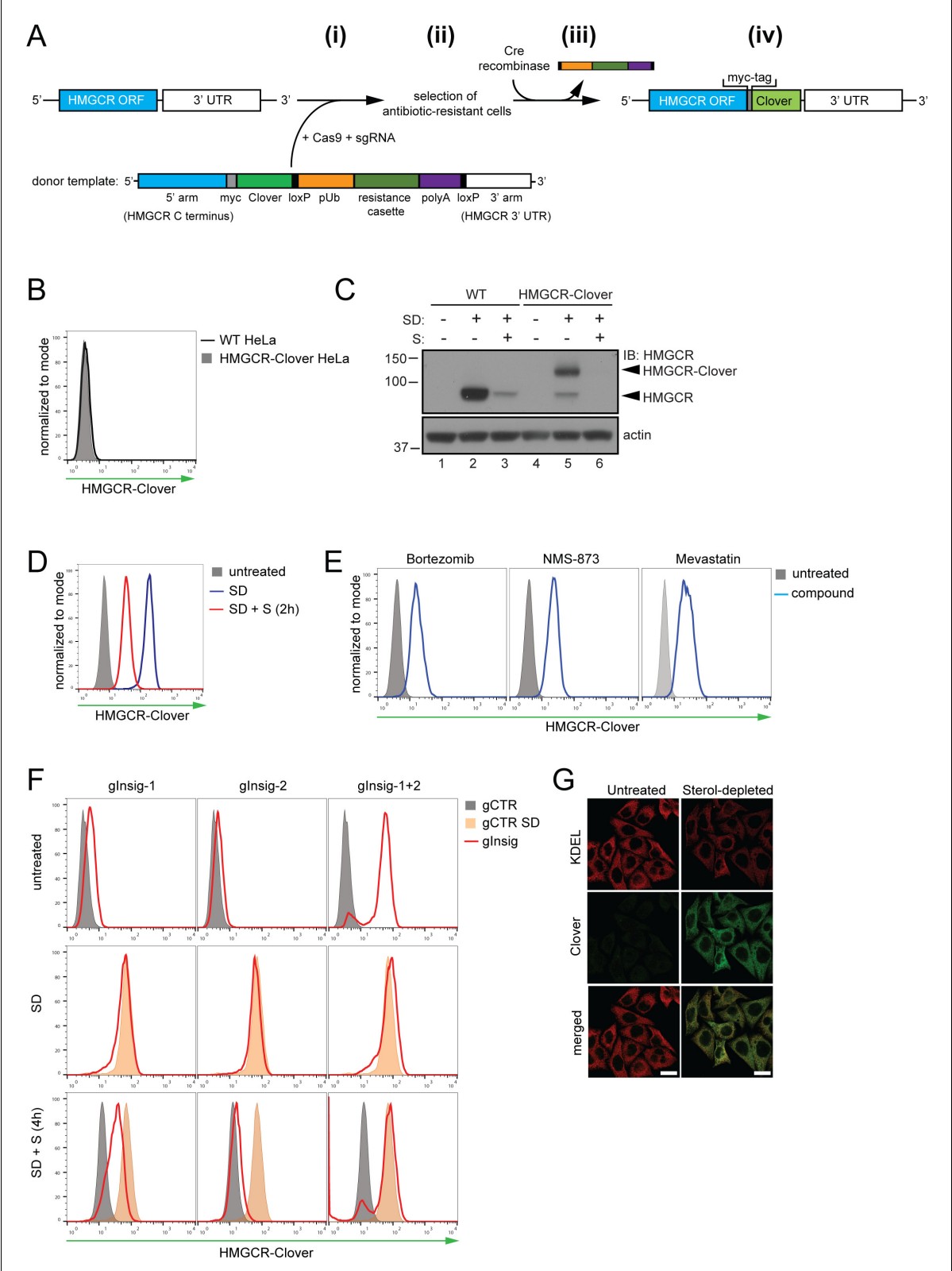

**Figure 1.** Fluorescent protein tagging of endogenous HMGCR generates a cholesterol-sensitive dynamic reporter. (**A**) Schematic showing generation of the HMGCR-Clover reporter. (i) The endogenous HMGCR locus of HeLa cells was modified by transfection of Cas9, sgRNA and a donor template. The 5' and 3' arm of the donor template were designed as homologous sequences encoding the C-terminal region and 3' UTR of the HMGCR gene. The C-terminal Clover tag (green) was appended in frame to the ORF of HMGCR (blue) including a myc-tag (grey) as spacer and an antibiotic

*Figure 1 continued on next page*

*Figure 1 continued*

resistance cassette flanked by loxP sites. (ii) Cells having stably integrated the recombination construct were enriched by antibiotic selection. (iii) The resistance cassette was removed by transient transfection of Cre recombinase to yield endogenous, C-terminally modified HMGCR (iv). pUb, ubiquitin promoter; ORF, open reading frame; UTR, untranslated region. (B – E) The HMGCR-Clover reporter phenocopies untagged HMGCR. (B) HMGCR-Clover expression (grey shaded histogram) as detected by flow cytometry under sterol-replete conditions. (C) Immunoblot of HMGCR in sterol-depleted (SD) HeLa WT *vs.* HMGCR-Clover cells -/+sterols (S) for 2 hr. For sterol depletion, cells were switched to SD medium (10% LPDS, mevastatin (10 µM), mevalonate (50 µM)) for 16 hr. Whole-cell lysates were separated by SDS-PAGE and HMGCR(-Clover) detected with an HMGCR-specific antibody. (D) Cytofluorometric analysis of HMGCR-Clover HeLa cells cultured in sterol-replete (shaded histogram) *vs.* sterol-depleted medium (SD) (16 hr, blue line histogram). Sterols (S) (2 µg/ml 25-hydroxycholesterol, 20 µg/ml cholesterol) were added back for 2 hr (red line histogram). (E) Flow cytometric analysis of HMGCR-Clover cells treated overnight with Bortezomib (25 nM), mevastatin (10 µM), or NMS-873 (10 µM) for 8 hr. (F) CRISPR/Cas9-mediated depletion of Insig-1 and −2 together (red line histogram) induce a dramatic increase in HMGCR-Clover expression, equivalent to sterol depletion (SD, orange shaded). HMGCR-Clover cells transiently expressing the indicated Insig-1/2 specific sgRNAs (four sgRNAs per gene) were treated as in (D) and, where indicated, sterols (S) added back for 4 hr (SD + S, bottom row). Representative of ≥3 independent experiments. (G) Immunofluorescence analysis of HMGCR-Clover and KDEL (ER marker) expression, showing co-localisation in sterol-depleted (SD, 16 hr) HMGCR-Clover HeLa cells. Scale bar = 20 µm.

DOI: https://doi.org/10.7554/eLife.40009.003

The following figure supplement is available for figure 1:

**Figure supplement 1.** Genotyping of HMGCR-Clover knock-in cells.

DOI: https://doi.org/10.7554/eLife.40009.004

of HMGCR. Inhibiting the enzymatic activity of HMGCR with mevastatin also stabilised HMGCR-Clover expression, as did inhibition of the proteasome (bortezomib) or p97/VCP (NMS-873) (*Figure 1E*), confirming the rapid, steady-state degradation of the HMGCR reporter. Furthermore, we showed that CRISPR/Cas9-mediated ablation of both Insig-1 and −2 together induced a dramatic increase in HMGCR-Clover expression, equivalent to levels seen following sterol depletion (*Figure 1F*). Under these conditions, the SREBP-SCAP complex is not retained in the ER, leading to constitutive SREBP-mediated transcription of HMGCR-Clover, irrespective of the sterol environment. CRISPR-mediated gene disruption of either Insig-1 or −2 alone caused only a small, steady-state rescue of HMGCR-Clover (*Figure 1F*), which was more pronounced with the loss of Insig-1 than Insig-2. While Insig-1-deficient cells were unable to completely degrade HMGCR upon sterol addition, only a minor defect in HMGCR degradation was seen in the absence of Insig-2 (*Figure 1F*), suggesting that Insig-1 is dominant over Insig-2 under these conditions. Finally, we confirmed that HMGCR-Clover was appropriately localised to the ER by confocal microscopy (*Figure 1G*). Thus, HMGCR-Clover is a dynamic, cholesterol-sensitive reporter, which rapidly responds to changes in intracellular cholesterol and is regulated in a proteasome-dependent manner.

## A genome-wide CRISPR/Cas9 screen identifies RNF145 as an E3 ligase required for HMGCR degradation

To identify genes required for the sterol-induced degradation of HMGCR, we performed a genome-wide CRISPR/Cas9 knockout screen in HMGCR-Clover cells. We took advantage of the rapid decrease in HMGCR-Clover expression following sterol addition to cells starved overnight (16 hr) of sterols (*Figure 1D*), and enriched for rare genetic mutants with reduced ability to degrade HMGCR-Clover in response to sterols. To this end, HMGCR-Clover cells were transduced with a genome-wide CRISPR/Cas9 knockout library comprising 10 sgRNAs per gene (*Morgens et al., 2017*). Mutagenised cells were first depleted of sterols overnight; sterols were then reintroduced for 5 hr, at which point rare mutant cells with reduced ability to degrade HMGCR-Clover upon sterol repletion were enriched by fluorescence-activated cell sorting (FACS) (*Figure 2A*, gating shown in *Figure 2—figure supplement 1A*). This process was repeated eight days later to further purify the selected cells. The enriched population contained only a small percentage of cells (1.96% after sort #1, 24.49% after sort #2) with increased steady-state HMGCR-Clover expression (green filled histogram in *Figure 2B*). However, the majority of sterol-starved cells from this selected population showed impaired degradation of HMGCR-Clover after addition of sterols (compare red *versus* orange filled histogram (*Figure 2B*, compare lanes 6 and 9 in *Figure 2—figure supplement 1B*)). The broad distribution of this histogram (*Figure 2B* red histogram) suggested that the enriched cell population contains a variety of mutants which differ in their ability to degrade HMGCR-Clover.

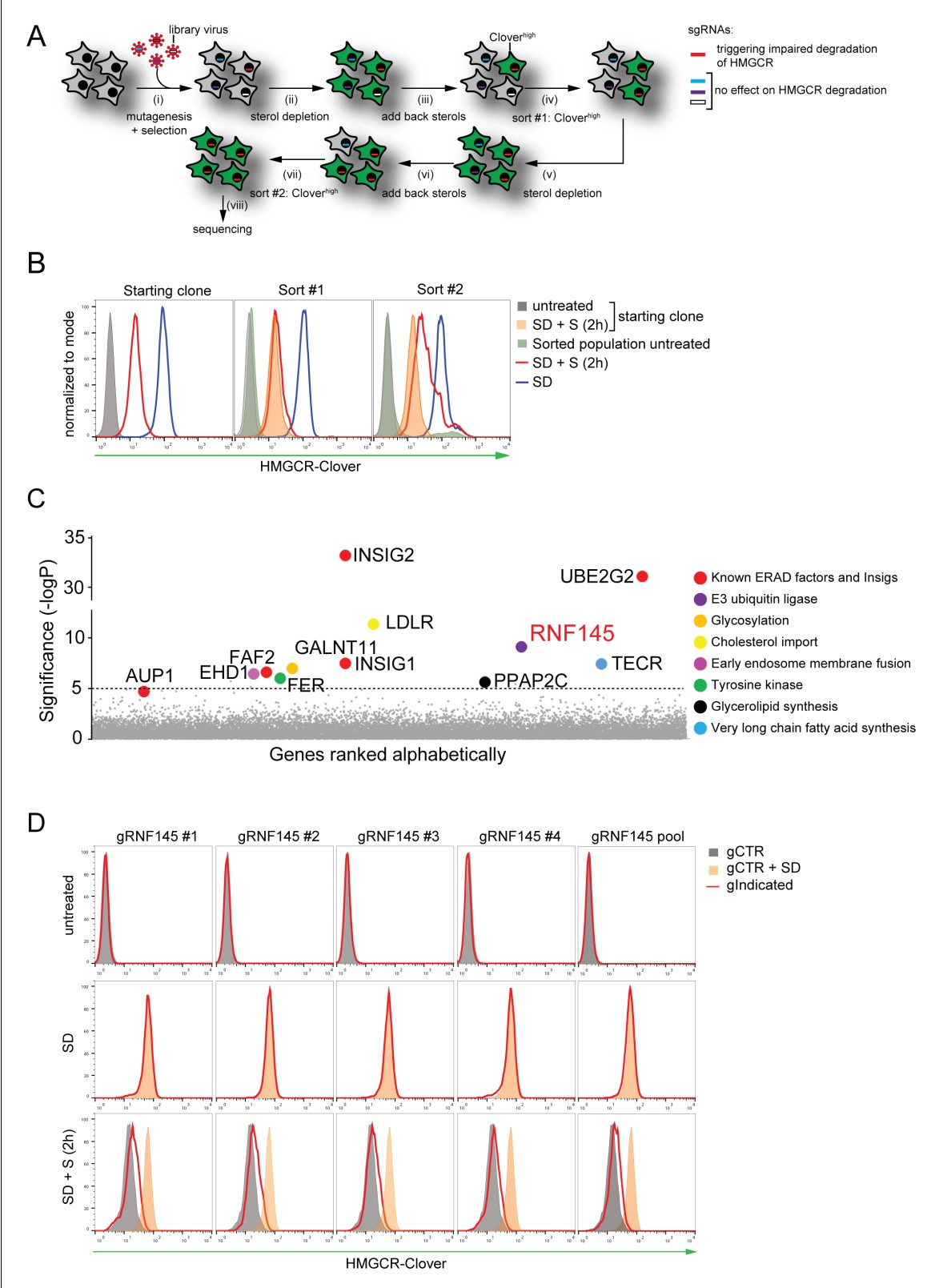

**Figure 2.** Genome-wide CRISPR knockout screen identifies a role for RNF145 in the sterol-dependent degradation of HMGCR. (A - B) Schematic view of the CRISPR/Cas9 knockout screen workflow and FACS enrichment. (**A**) HMGCR-Clover HeLa cells transduced with a genome-wide sgRNA library targeting 19930 genes (i) were subjected to overnight sterol-starvation followed by sterol repletion for 5 hr (ii, iii, v, vi). Mutants unable to degrade HMGCR-Clover despite sterol repletion (Clover[high]) were enriched by two sequential rounds of FACS (iv, vii) and candidate genes identified by deep

*Figure 2 continued on next page*

*Figure 2 continued*

sequencing (viii). **(B)** Enrichment of HMGCR-Clover mutants after sort #1 and sort #2 (red line histograms, corresponding to steps 'iv' and 'vii' in *Figure 2A*) as determined by flow cytometry. Cells were treated as described in *Figure 1D*. SD, sterol-depleted; S, sterols. **(C)** Candidate genes identified in the genome-wide knockout screen. Genes scoring above the significance threshold of - logP ≥5 (dotted line) and AUP1 (- logP = 4.7) are highlighted. **(D)** RNF145 depletion mildly impairs sterol-accelerated HMGCR-Clover degradation. HMGCR-Clover cells transiently expressing four independent RNF145-specific sgRNAs (gRNF145#1–4, red line histogram), individually or as a pool, *vs.* sgB2M (gCTR) were sterol-depleted overnight (middle row, SD) and re-examined by flow cytometry following 2 hr sterol addition (bottom row, SD + S). Representative of ≥3 independent experiments.

DOI: https://doi.org/10.7554/eLife.40009.005

The following figure supplement is available for figure 2:

**Figure supplement 1.** Genome-wide screen for proteins involved in HMGCR degradation.

DOI: https://doi.org/10.7554/eLife.40009.006

The sgRNAs in the selected cells, and an unselected control library, were sequenced on the Illumina HiSeq platform (*Figure 2A* (viii)). Using the RSA algorithm, we identified a set of 11 genes, which showed significant enrichment (-logP >5) in the selected cells (*Figure 2C*). Many of these are known to be required for the sterol-induced degradation of HMGCR (*König et al., 2007*). The screen identified the E2 ubiquitin conjugating enzyme UBE2G2 and its accessory factor AUP1, which recruits UBE2G2 to lipid droplets and membrane E3 ubiquitin ligases (*Klemm et al., 2011*; *Jo et al., 2013*; *Spandl et al., 2011*; *Christianson et al., 2011*), as well as both Insig-1 and −2 (*Yabe et al., 2002*; *Yang et al., 2002*; *Sever et al., 2003a*). The role of the remaining hits is summarized (*Table 1*) and validation of selected hits is shown (Insig-1/2, *Figure 1F*; UBE2G2, EHD1, GALNT11, LDLR and TECR, *Figure 2 – figure supplement 1C/D*).

Strikingly, the only ER-resident E3 ubiquitin ligase to emerge from the screen is the poorly characterised RNF145. RNF145 shares 27% amino acid identity with TRC8, which is one of the E3 ligases (together with gp78) previously suggested to ubiquitinate HMGCR (*Jo et al., 2011*). Interestingly, RNF145 also harbours a YLYF motif at its N-terminus, which is similar to the YIYF motif present in the sterol-sensing domain (SSD) of SCAP and HMGCR required for their binding to the Insig proteins (*Yang et al., 2002*; *Sever et al., 2003a*; *Jiang et al., 2018*; *Cook et al., 2017*; *Zhang et al., 2017*). The presence of the YLYF motif suggested that RNF145 might itself interact with the Insig proteins and therefore represented a promising candidate from our genetic screen.

To see if we could validate the role of RNF145 in HMGCR degradation, we designed four independent sgRNAs, either targeting RNF145 individually or as a pool. Under cholesterol-replete conditions, no accumulation of the HMGCR-reporter was observed in RNF145-depleted cells (top and

**Table 1.** Candidate genes (- log(p)≥5) identified in a genome-wide CRISPR/Cas9 screen for proteins involved HMGCR degradation.

| Gene | Full name | -log(p)* | Function |
|---|---|---|---|
| AUP1 | Ancient Ubiquitous Protein 1 | 4.70 | ERAD |
| EHD1 | EH Domain Containing 1 | 6.50 | Early endosome membrane fusion |
| FAF2 | Fas Associated Factor Family Member 2 | 6.63 | ERAD |
| FER | Tyrosine-protein Kinase Fer | 6.05 | Tyrosine kinase |
| GALNT11 | Polypeptide N-acetylgalactosaminyltransferase 11 | 7.03 | Protein glycosylation |
| INSIG1 | Insulin Induced Gene 1 | 7.50 | Cholesterol metabolism |
| INSIG2 | Insulin Induced Gene 2 | 33.24 | Cholesterol metabolism |
| LDLR | Low Density Lipoprotein Receptor | 11.44 | Cholesterol metabolism |
| PPAP2C | Phospholipid Phosphatase 2 | 5.67 | Glycerolipid synthesis |
| RNF145 | RING Finger Protein 145 | 9.18 | E3 ubiquitin ligase |
| TECR | Trans-2,3-enoyl CoA Reductase | 7.45 | Very-long chain fatty acid synthesis |
| UBE2G2 | Ubiquitin Conjugating Enzyme E2 G2 | 31.14 | E2 ubiquitin conjugating enzyme |

*Only statistically significant hits (-log(p)≥5) are shown.

DOI: https://doi.org/10.7554/eLife.40009.007

middle rows, *Figure 2D*), but a small and highly reproducible decrease in HMGCR-Clover degradation was seen following re-introduction of sterols (red histograms, bottom row in *Figure 2D*), emphasising the utility of the endogenous fluorescent reporter in identifying subtle phenotypes. Since the identity of the E3 ubiquitin ligases regulating HMGCR turnover remains controversial, the modest effect of RNF145 loss on HMGCR-Clover sterol-induced degradation suggested the involvement of additional ligase(s). Our screen therefore identified both known and novel components implicated in sterol-dependent HMGCR ERAD.

## RNF145 together with gp78 are required for HMGCR degradation

If a second E3 ligase is partially redundant with RNF145, its effect should be unmasked in RNF145-deficient cells. We therefore generated a focussed subgenomic sgRNA library targeting 1119 genes of the ubiquitin-proteasome system as described in 'Materials and methods', including 830 predicted E3 ubiquitin ligases, and used this library to screen for genes required for the degradation of HMGCR in RNF145-deficient HMGCR-Clover cells (*Figure 3—figure supplement 4B*, lane two for knockout validation). Due to the reduced complexity of this focussed library, only a single FACS enrichment step was used (*Figure 3A*, red histogram).

Strikingly, this screen identified gp78 (gene name: *AMFR*) (*Figure 3B*, *Table 2*), the E3 ubiquitin ligase previously implicated in HMGCR degradation (*Jo et al., 2011*; *Song et al., 2005*; *Fang et al., 2001*). Taking a combined knockout strategy we asked whether gp78 and RNF145 are together responsible for HMGCR degradation. As predicted by the genetic approach (*Figure 3C* **(ii)**), there was no difference in sterol-induced HMGCR-Clover degradation between control and gp78-depleted HMGCR-Clover cells. Gp78 was not, therefore, a false-negative from our initial, genome-wide CRISPR/Cas9 screen (*Figure 2C*). Individual knockout of RNF145 again showed that sterol-induced HMGCR-Clover degradation was mildly impaired in RNF145-depleted cells (*Figure 3C* (iii)). However, sgRNA-mediated targeting of gp78 together with RNF145 (*Figure 3C* (iv), see *Figure 3—figure supplement 4A and B* lane three for knockout validation), resulted in a significant increase in both steady-state HMGCR-Clover (*Figure 3C* (iv) grey to green filled histograms) and an inability to degrade HMGCR-Clover upon addition of sterols to sterol-starved cells (*Figure 3C* (iv) blue to red histogram), a phenotype comparable to UBE2G2 deletion (*Figure 3C* (v)). Our results therefore suggest a partial functional redundancy between gp78 and RNF145 and imply that both ligases can independently regulate the sterol-induced degradation of HMGCR.

## RNF145 and gp78 regulate endogenous wild type HMGCR

To confirm that the phenotypes observed in RNF145- and gp78-deficient HMGCR-Clover cells were representative of endogenous, wild type HMGCR regulation, we deleted RNF145 and/or gp78 from WT HeLa cells and monitored endogenous HMGCR by immunoblot analysis. The sterol-induced degradation of HMGCR was assessed in four RNF145 knockout clones, derived from two different sgRNAs (validation in *Figure 3—figure supplement 1A,B*). No difference in the sterol-induced degradation of HMGCR was seen in these RNF145 knockout clones (*Figure 3—figure supplement 2*, compare lanes 6 and 7–10). The subtle effect on HMGCR-Clover expression revealed by flow cytometry (*Figures 2D* and *3C*) may not be detected by the less sensitive immunoblot analysis. Similarly, loss of gp78 alone (*Figure 3—figure supplement 3A* for sgRNA validation) did not affect HMGCR degradation (*Figure 3D*, compare lanes 6 and 7–10), but loss of gp78 together with RNF145 resulted in a significant rescue of steady state HMGCR (*Figure 3E*, *Figure 3—figure supplement 3C*). Following sterol addition, gp78/RNF145 double-knockout clones showed a marked (although still incomplete) reduction in sterol-induced HMGCR degradation (*Figure 3F*, compare lanes 7 + 8 with 9 –12). These data validate the phenotypes exhibited by the HMGCR-Clover reporter cell line and confirm a role for both gp78 and RNF145 in the sterol-induced degradation of endogenous HMGCR.

## RNF145 E3 ubiquitin ligase activity is required for HMGCR degradation

To determine whether RNF145 E3 ubiquitin ligase activity is required for HMGCR degradation, we complemented a population of gp78/RNF145 double-knockout HMGCR-Clover cells (*Figure 3—figure supplement 4B*, lane three for knockout validation) with either epitope-tagged wild type RNF145, or a catalytically-inactive RNF145 RING domain mutant (C552A, H554A) (*Figure 4A*). The

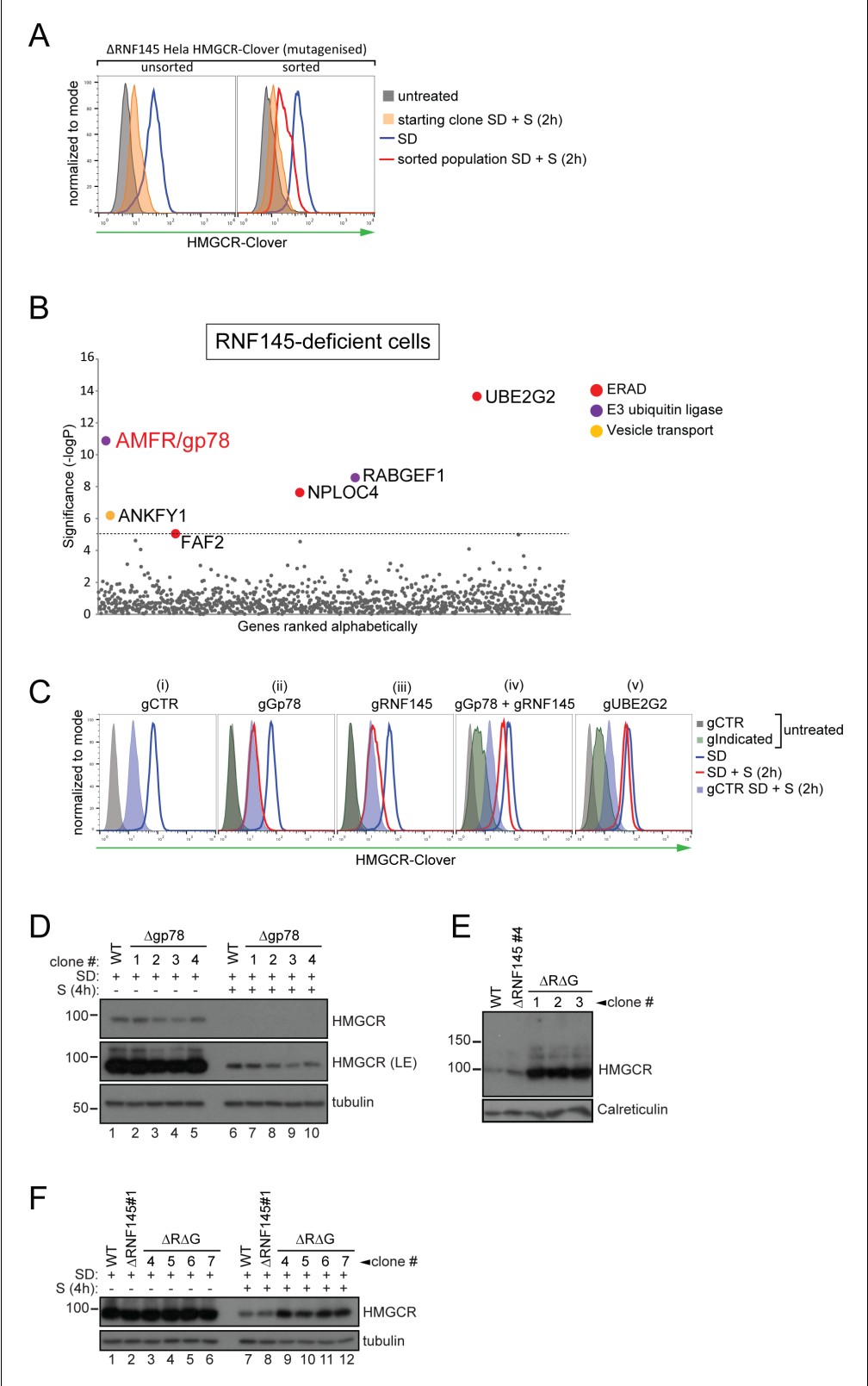

**Figure 3.** RNF145 together with gp78 are required for HMGCR degradation. (A - B) FACS enrichment and scatter plot of candidate genes identified in the ubiquitome-targeted knockout screen. (**A**) HMGCR-Clover ΔRNF145#5 HeLa cells were mutagenized using a targeted ubiquitome-specific sgRNA library and mutant cells showing impaired sterol-dependent degradation of HMGCR-Clover were enriched by FACS. Enrichment is represented by a broad population of Clover[high] cells in the presence of sterols (S, 2 hr) after overnight sterol depletion (SD, blue to red histogram). (**B**) Genes scoring

*Figure 3 continued on next page*

*Figure 3 continued*

above the significance threshold of - logP ≥5 (dotted line) are highlighted. (**C - F**) sgRNA targeting of gp78 together with RNF145 increases steady-state HMGCR-Clover and inhibits sterol-accelerated degradation of HMGCR-Clover in sterol-starved cells. (**C**) HMGCR-Clover cells transiently transfected with indicated sgRNAs were sterol-depleted (SD) overnight (blue line histogram) and sterols (2 µg/ml 25-hydroxycholesterol, 20 µg/ml cholesterol) added back (SD + S) for 2 hr (red line histogram or blue shaded histogram for sgB2M (gCTR)). Representative of ≥3 independent experiments. (**D**) Four independent gp78 knockout clones (#1–4) or WT cells were sterol-depleted (16 hr) ± S (4 hr) and HMGCR levels monitored by immunoblotting. LE, long exposure. (**E**) HMGCR steady-state levels in three RNF145/gp78 double knockout clones (ΔRΔG #1–3). (**F**) Four RNF145/gp78 double-knockout clones (ΔRΔG #4–7), RNF145 knockout, and WT cells were sterol-depleted (SD) overnight and HMGCR expression assessed ± sterols (S, 4 hr) by immunoblot analysis.

DOI: https://doi.org/10.7554/eLife.40009.008

The following figure supplements are available for figure 3:

**Figure supplement 1.** Validation of RNF145 knockout clones.

DOI: https://doi.org/10.7554/eLife.40009.009

**Figure supplement 2.** RNF145 loss is insufficient to block sterol-induced HMGCR degradation.

DOI: https://doi.org/10.7554/eLife.40009.010

**Figure supplement 3.** RNF145/gp78 double-knockout cells show increased HMGCR at steady-state and impaired sterol-induced HMGCR degradation.

DOI: https://doi.org/10.7554/eLife.40009.011

**Figure supplement 4.** Establishment of RNF145/gp78 knockout HeLa HMCR-Clover and RNF145 complementation cell lines.

DOI: https://doi.org/10.7554/eLife.40009.012

pronounced block in the sterol-induced degradation of HMGCR-Clover was at least partially rescued by expression of wild type, but not the RNF145 RING domain mutant (*Figure 4B*, compare blue to red histogram). The E3 ligase activity of RNF145 is therefore critical for HMGCR ERAD.

## Endogenous RNF145 is an unstable E3 ligase, whose transcription is sterol-regulated

Endogenous RNF145 has a short half-life (~2 hr) and displayed rapid, proteasome-mediated degradation (*Figure 5A* (i)), an observation confirmed in multiple cell lines (*Figure 5—figure supplement 1A*). This rapid turnover of endogenous RNF145 contrasts sharply with the stability of endogenous gp78, which shows little degradation over the 10 hr chase period (*Figure 5A* (i)). Although RNF145 and gp78 both target HMGCR for degradation, the two ligases did not appear to be co-regulated as RNF145 stability was unaffected by gp78 and vice-versa (*Figure 5A* (i, ii), *Figure 5—figure supplement 1B*). However, endogenous RNF145 was stabilised by deletion of its cognate E2 enzyme UBE2G2 (*Figure 5B*), and, furthermore, the catalytically-inactive RING domain mutant expressed in RNF145-deficient cells (ΔRNF145 #4 + RNF145-V5 (mut)) exhibited greater abundance at steady-state compared with its wild type counterpart (*Figure 3—figure supplement 4C*). Together these data show that RNF145 is intrinsically unstable and rapidly turned over in an auto-regulatory manner.

Since RNF145 is rapidly turned over, we aimed to determine whether RNF145 gene transcription was sterol-responsive. Sterol depletion induced RNF145 (~2.99 ± 0.65 fold increase, p=0.0009) mRNA expression as well as HMGCR (~12.26 ± 3.16 fold increase, p=0.0004) mRNA expression

**Table 2.** Candidate genes (-log(p)≥5) identified in a ubiquitome CRISPR/Cas9 screen for proteins mediating HMGCR degradation in RNF145-deficient cells.

| Gene | Full name | -log(p)* | Function |
|------|-----------|----------|----------|
| AMFR | Gp78/Autocrine Motility Factor Receptor | 10.87 | E3 ubiquitin ligase |
| ANKFY1 | Ankyrin Repeat And FYVE Domain Containing 1 | 6.19 | Proposed Rab5 effector |
| FAF2 | Fas Associated Factor Family Member 2 | 5.05 | ERAD |
| NPLOC4 | NPL4 Homolog | 7.63 | Ubiquitin recognition factor |
| RABGEF1 | RAB Guanine Nucleotide Exchange Factor 1 | 8.56 | Nucleotide exchange factor, E3 ubiquitin ligase |
| UBE2G2 | Ubiquitin Conjugating Enzyme E2 G2 | 13.66 | E2 ubiquitin conjugating enzyme |

*Only statistically significant hits (-log(p)≥5) are shown.

DOI: https://doi.org/10.7554/eLife.40009.013

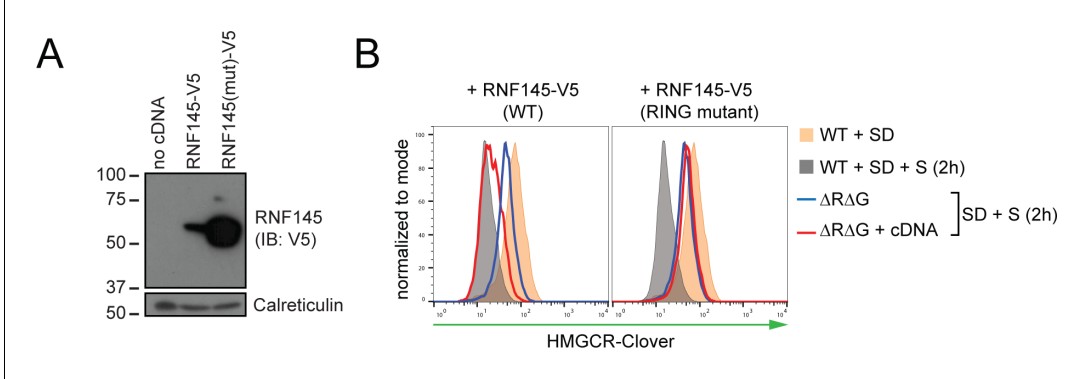

**Figure 4.** RNF145 E3 ligase activity is required for HMGCR degradation. (**A**) Exogenous expression of RNF145 and RING-mutant RNF145 in HMGCR-Clover HeLa cells. RNF145/gp78 double-knockout HMGCR-Clover cells were transduced with lentivirus expressing either RNF145-V5 (WT) or a catalytically inactive RING domain mutant RNF145(C552A, H554A)-V5 cDNA (RING mutant) and cell lysates separated by SDS-PAGE and visualised by immunoblot analysis. IB, immunoblot. (**B**) Wild type, but not RING mutant RNF145, complements the RNF145-deficiency phenotype. RNF145/gp78 double-knockout HMGCR-Clover cells (ΔRΔG #11) were transduced with lentivirus expressing either RNF145-V5 or a catalytically inactive RING domain mutant RNF145(C552A, H554A)-V5 cDNA. Cells were sterol-depleted (16 hr) and after sterol repletion (2 hr), HMGCR-Clover levels were assessed by flow cytometry.
DOI: https://doi.org/10.7554/eLife.40009.014

(*Figure 5C*). This accumulation of endogenous RNF145 was suppressed following the addition of methyl beta-cyclodextrin (MBCD)-complexed cholesterol (Chol/MBCD) to the starvation media (*Figure 5D*), whereas gp78 abundance remained unaltered (*Figure 5—figure supplement 1D*). RNF145 is therefore a unique, sterol-regulated E3 ubiquitin ligase whose expression is dependent on the cellular sterol status.

## Endogenous RNF145 shows a sterol-sensitive interaction with HMGCR and Insig-1

The Insig proteins provide an ER-resident platform for sterol-dependent interactions between HMGCR and its regulatory components (*Dong et al., 2012*). Since RNF145 is sterol-regulated and degrades HMGCR we initially wanted to know if RNF145 interacts with HMGCR. We found that in sterol-replete but not sterol-deplete conditions, endogenous HMGCR co-immunoprecipitates both epitope-tagged RNF145 (*Figure 6A*, *Figure 3—figure supplement 4C* lane three for relative RNF145-V5 levels upon reconstitution), as well as endogenous RNF145 (*Figure 6B*). Initial attempts to ascertain whether this interaction between RNF145 and HMGCR was direct, or mediated *via* the Insig proteins were challenging due to the low expression levels of endogenous RNF145. We circumvented this problem by performing the co-immunoprecipitation in UBE2G2 knockout cells, which express increased levels of endogenous RNF145 (*Figure 5B*). Under these conditions, RNF145 showed a clear, sterol-dependent interaction with Insig-1, correlating with RNF145's association with HMGCR (*Figure 6C*). Importantly, endogenous RNF145 is not, therefore, continually bound to Insig-1, but, like HMGCR, associates with Insig-1 in a sterol-dependent manner.

Binding of RNF145 to Insig-1 was HMGCR-independent (*Figure 6D*) and, in the absence of Insigs, RNF145 was unable to bind HMGCR (*Figure 6E*, see *Figure 6—figure supplement 1A–C* for generation of Insig-1+2 knockout cells). Insigs are therefore indispensable for the interaction between RNF145 and HMGCR. These findings emphasize the central role of Insig proteins as scaffolds in the sterol-induced engagement of HMGCR by RNF145.

## In the absence of RNF145 and gp78, Hrd1 targets HMGCR for degradation

Despite our two genetic screens identifying a requirement for RNF145 and gp78 in HMGCR degradation (*Figures 2C* and *3B*), the combined loss of these two ligases failed to completely inhibit sterol-induced HMGCR degradation (*Figure 3C* (iv); *Figure 7A* (ii)). Furthermore, ablation of UBE2G2 in RNF145/gp78 double-knockout cells exacerbated the sterol-dependent degradation defect

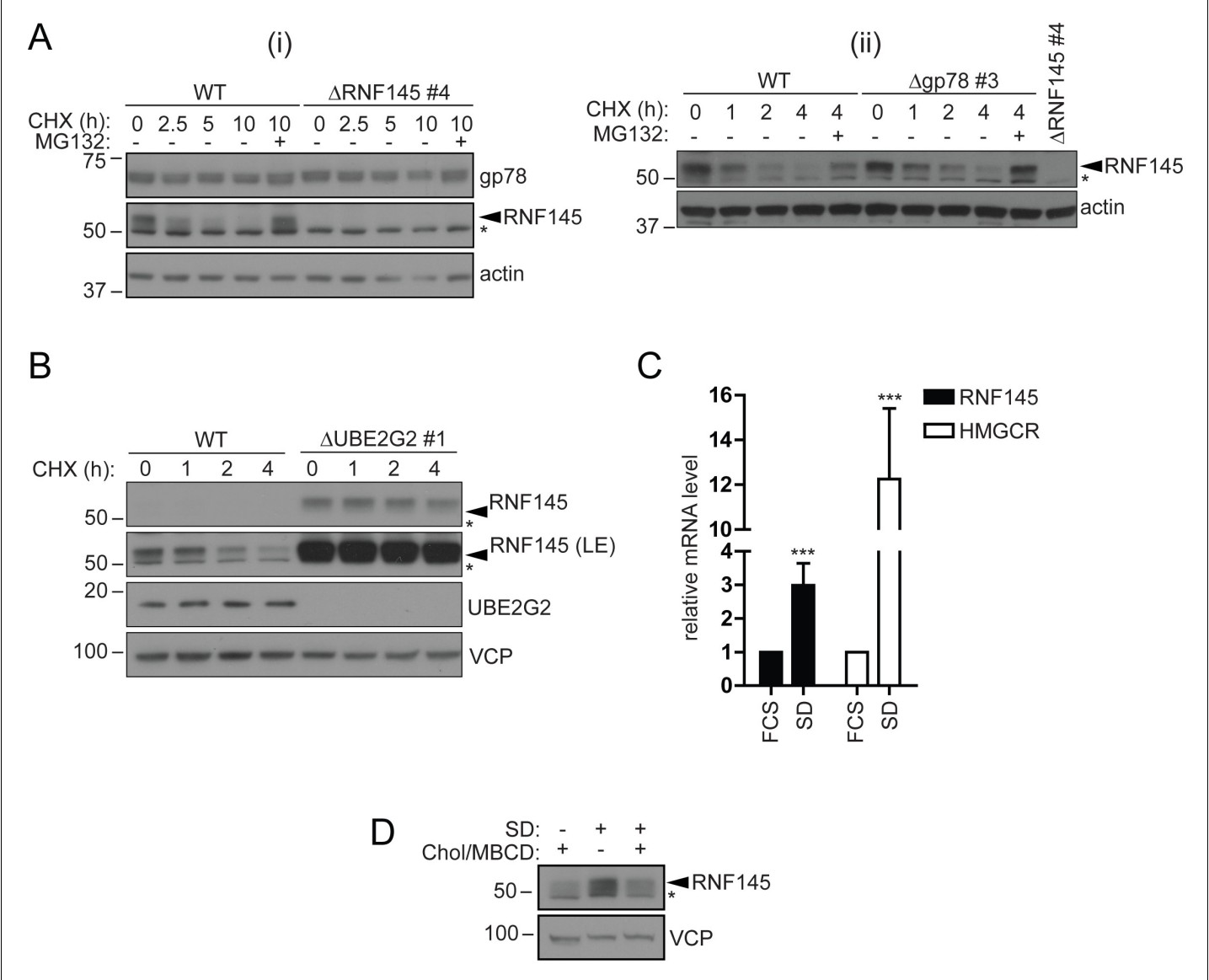

**Figure 5.** RNF145 is an intrinsically unstable, sterol-responsive E3 ligase. (**A and B**) RNF145 has a short half-life and is auto-regulated by UBE2G2. (**A**) Translational shutoff analysis of gp78 in WT *versus* ΔRNF145 #4 (i) or RNF145 in Δgp78 #3 cells (ii) treated with cycloheximide (CHX, 1 µg/ml) ± MG132 (20 µg/ml) for the indicated times. Non-specific bands are indicated by an asterisk (*). Representative of ≥2 independent experiments. (**B**) Immunoblot analysis of WT and ΔUBE2G2 HeLa cells treated with CHX (1 µg/ml) for the indicated times. VCP serves as a loading control. LE, long exposure. An asterisk (*) signifies non-specific bands. (**C and D**) Sterol depletion induces transcriptional activation and increased levels of RNF145 protein. (**C**) Relative RNF145 and HMGCR mRNA levels as measured by quantitative PCR in HeLa cells grown in 10% FCS (FCS) or sterol-depleted (SD, 10% LPDS + 10 µM mevastatin + 50 µM mevalonate) for 48 hr. Mean ± S.D. (n = 4) and significance are shown, unpaired Students t-test: ***p≤0.001. (**D**) HeLa cells were grown under sterol-rich or -deplete conditions (±SD, as indicated) for 48 hr in the presence of mevastatin (10 µM) and mevalonate (50 µM) ± complexed cholesterol (Chol:MBCD, 37.5 µM). Whole-cell lysates were separated by SDS-PAGE and underwent immunoblot analysis. Non-specific bands are indicated (*). Representative of ≥2 independent experiments.

DOI: https://doi.org/10.7554/eLife.40009.015

The following source data and figure supplements are available for figure 5:

**Source data 1.** Raw data from qPCR experiment in *Figure 5C*.

DOI: https://doi.org/10.7554/eLife.40009.019

**Figure supplement 1.** RNF145 is rapidly degraded by the ubiquitin proteasome system.

DOI: https://doi.org/10.7554/eLife.40009.016

**Figure supplement 2.** Increased RNF145 transcription upon sterol depletion is LXR-independent.

DOI: https://doi.org/10.7554/eLife.40009.017

*Figure 5 continued on next page*

*Figure 5 continued*

**Figure supplement 2—source data 1.** Raw data from qPCR experiment in *Figure 5—figure supplement 2B*.
DOI: https://doi.org/10.7554/eLife.40009.018

(*Figure 7A* (iv)), predicting the role for an additional E3 ubiquitin ligase(s) utilising UBE2G2 in HMGCR degradation. We therefore assessed whether ablation of either of the two remaining ER-resident E3 ligases known to use UBE2G2, TRC8 (*van de Weijer et al., 2017*) and Hrd1 (*Kikkert et al., 2004*), exacerbated the HMGCR-degradation defect in RNF145/gp78 double-knock-out cells (*Figure 7B*, *Figure 7—figure supplements 1B* and *2B* for knockdown validation). While the loss of TRC8 had no effect on HMGCR-Clover expression, the loss of Hrd1 in RNF145/gp78 double-knockout cells increased steady-state HMGCR-Clover expression and caused a complete block in the sterol-accelerated degradation of HMGCR-Clover (*Figure 7B* (ii), *Figure 7—figure supplement 1A* for validation with individual independent sgRNAs). This additive effect of Hrd1 depletion on the sterol-induced turnover of endogenous HMGCR was independently confirmed by immunoblot analysis (*Figure 7C*, compare lanes 2, 4 and 6) and was observed as early as 60 min after sterol addition (*Figure 7—figure supplement 1D*, compare lanes 7 and 9). Importantly, depletion of Hrd1, alone or in combination with depletion of either gp78 or RNF145, did not affect HMGCR-Clover degradation (*Figure 7—figure supplement 1C*). Moreover, TRC8 depletion affected neither steady-state HMGCR-Clover expression, nor sterol-induced HMGCR-Clover degradation (*Figure 7B* (iii)). Indeed, despite a functional TRC8 depletion (*Figure 7—figure supplement 2B* for validation of TRC8 depletion) (*Stagg et al., 2009*), we could detect no role for TRC8, depleted either alone or in combination with RNF145, in the sterol-induced degradation of HMGCR (*Figure 7—figure supplement 2A*).

In summary, gp78 with RNF145 are the only combination of ligases whose loss inhibited HMGCR degradation. Hrd1 depletion also delays sterol-induced HMGCR degradation, but only in the absence of RNF145 and gp78.

## RNF145, gp78 and Hrd1 are required for sterol-accelerated HMGCR ubiquitination

As a complete block of sterol-accelerated HMGCR degradation required the depletion of all three UBE2G2-dependent E3 ubiquitin ligases, we determined how the sequential depletion of these ligases affected the ubiquitination status of HMGCR. The combined loss of RNF145 with gp78 showed a dramatic reduction in HMGCR ubiquitination, but a complete loss of ubiquitination required the depletion of all three ligases (*Figure 7D*). As predicted, depletion of UBE2G2 also caused a marked decrease in HMGCR ubiquitination. Taken together, these results demonstrate the remarkable plasticity of the HMGCR-degradation machinery.

## Discussion

The generation of a dynamic, cholesterol-sensitive endogenous HMGCR reporter cell line allowed an unbiased genetic approach to identify the cellular machinery required for sterol-accelerated HMGCR degradation. This reporter cell line has the advantage of being able to identify both complete and partial phenotypes and helps explain why the identity of the E3 ubiquitin ligases responsible for the sterol-accelerated degradation of HMGCR has remained controversial. We find that three E3 ubiquitin ligases - RNF145, gp78 and Hrd1 - are together responsible for HMGCR degradation (*Figure 8*). The activity of the two primary ligases, RNF145 and gp78 is partially redundant as the loss of gp78 alone did not affect HMGCR degradation, while loss of RNF145 showed only a small reduction on HMGCR degradation. In the absence of both RNF145 and gp78, a third ligase, Hrd1, can compensate and partially regulate HGMCR degradation, but this effect of Hrd1 is only revealed in the absence of both RNF145 and gp78, and in no other identified combination.

Initial reports of a role for gp78 in HMGCR degradation, either alone (*Song et al., 2005*) or in combination with TRC8 (*Jo et al., 2011*), were not reproduced in an independent study (*Tsai et al., 2012*) and so this important issue has remained unresolved. Our initial genome-wide screen successfully identified a single E3 ligase (RNF145) as well as many of the components known to be required for sterol-accelerated HMGCR degradation (e.g. Insig-1/2, UBE2G2, AUP1, FAF2; *Figure 2C*)

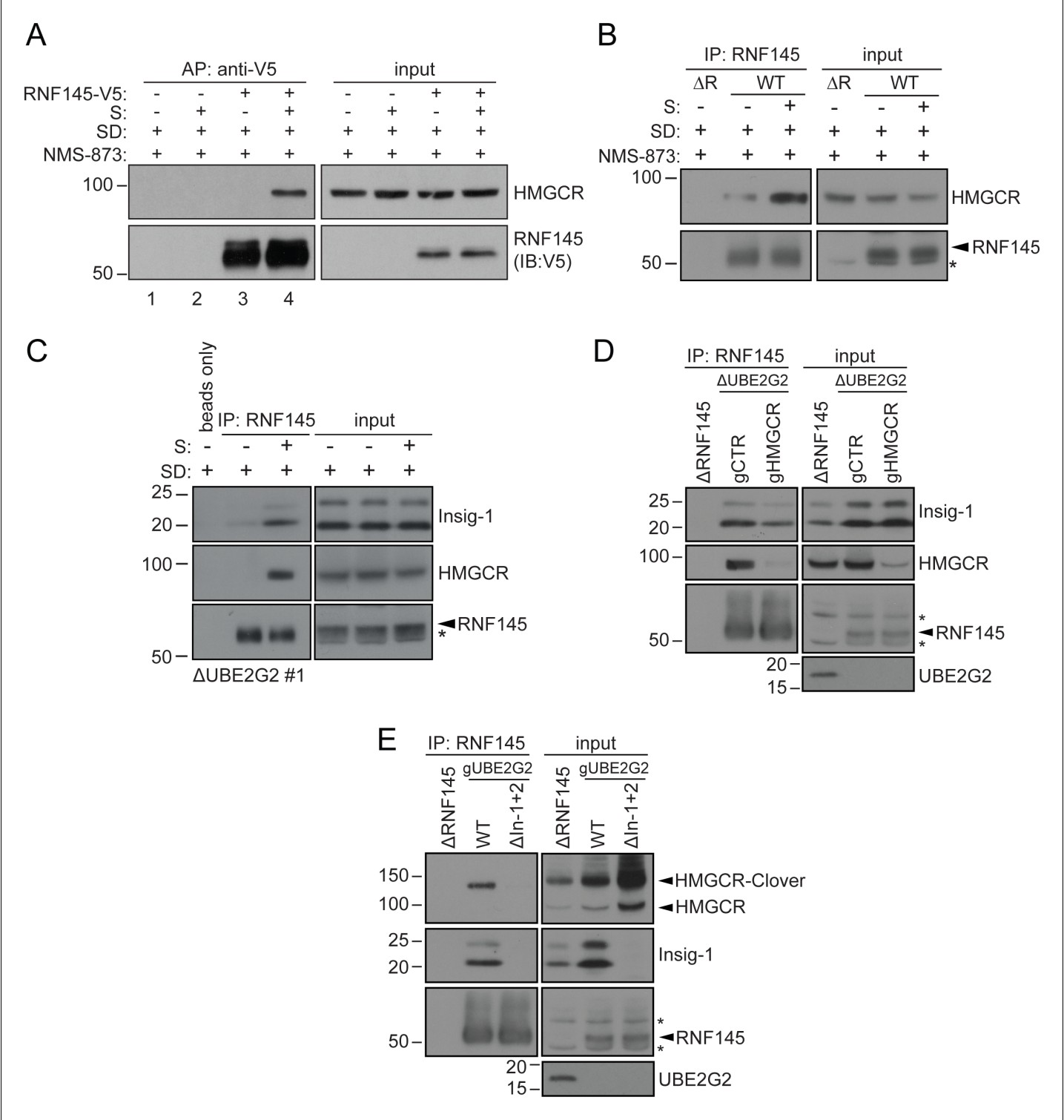

**Figure 6.** Endogenous RNF145 shows sterol-sensitive binding to Insig-1 and HMGCR. (**A**) Exogenous RNF145 shows sterol-sensitive binding to HMGCR. RNF145 knockout cells stably reconstituted with RNF145-V5 (ΔR145 #4 + R145-V5, as shown in *Figure 3—figure supplement 4C*, lane 3) were sterol-depleted (SD, 20 hr) and, where indicated, sterols (**S**) added back for 1 hr in the presence of NMS-873 (10 μM, 1.5 hr). RNF145-V5 was affinity-purified (AP) and HMGCR detected by immunoblotting. Representative of ≥3 independent experiments. (**B - C**) Endogenous RNF145 shows sterol-sensitive binding to HMGCR and Insig-1. (**B**) HeLa WT or ΔRNF145 #4 (ΔR) cells were treated as in (**A**), endogenous RNF145 was immunoprecipitated (IP), and RNF145 and HMGCR detected by immunoblot analysis. Non-specific bands are designated by an asterisk (*). Representative of ≥3 independent experiments. (**C**) HeLa UBE2G2 knockout cells (ΔUBE2G2 #1) were sterol-depleted (SD, 20 hr) and, where indicated, sterols (**S**) added for 1

*Figure 6 continued on next page*

*Figure 6 continued*

hr. Endogenous RNF145 was affinity-purified and following SDS-PAGE separation, Insig-1 and HMGCR detected by immunoblot analysis. Representative of ≥2 independent experiments. (D - E) Insigs mediate binding between RNF145 and HMGCR. (D) HeLa UBE2G2 knockout cells (ΔUBE2G2 #1) transfected with a pool of four sgRNAs targeting HMGCR (gHMGCR) or sgRNA targeting B2M (gCTR) were enriched by puromycin selection and sterol-depleted for 20 hr, before adding back sterols for 1 hr + NMS-873 (10 μM, 1.5 hr). Endogenous RNF145 was immunoprecipitated and Insig-1 and HMGCR detected by immunoblot analysis. Non-specific bands are designated by an asterisk (*). (E) HMGCR-Clover HeLa WT or Insig-1 +2 knockout (ΔIn-1+2) cells were transfected with a pool of three sgRNAs targeting UBE2G2 (gUBE2G2) and treated as in (D). Non-specific bands are designated by an asterisk (*).

DOI: https://doi.org/10.7554/eLife.40009.020

The following figure supplement is available for figure 6:

**Figure supplement 1.** Generation of Insig knockout cell lines.

DOI: https://doi.org/10.7554/eLife.40009.021

(*Sever et al., 2003b*; *Miao et al., 2010*; *Jo et al., 2013*), thus validating the suitability of this genetic approach.

For a small number of validated hits from our screen (*Figure 2C*, *Figure 2—figure supplement 1D*), the effects on sterol-accelerated HMGCR degradation were unanticipated and likely reflect wide-ranging alterations to the protein and lipid environment. Trans-2,3-enoyl CoA reductase (TECR) catalyses the final steps in the synthesis of very long-chain fatty acids (VLCFAs) (*Moon and Horton, 2003*) as well as the saturation step in sphingolipid degradation (*Wakashima et al., 2014*). Polypeptide N-acetylgalactosaminyltransferase 11 (GALNT11) initiates protein O-linked glycosylation, suggesting that a protein involved in HMGCR regulation requires O-linked glycosylation (*Schwientek et al., 2002*). Interestingly, our screen revealed that loss of the LDLR impaired HMGCR-Clover degradation. This finding is unexpected as the cholesterol added to the cells to induce HMGCR degradation was not in the form of LDL. EH domain-containing protein 1 (EHD1) is required for the internalisation and recycling of several plasma membrane receptors, including the LDLR (*Naslavsky et al., 2007*; *Naslavsky and Caplan, 2011*) and loss of EDH1 impairs LDLR trafficking with decreased intracellular cholesterol levels (*Naslavsky et al., 2007*). The other significant hits in the screen (PPAP2C, FER) have not been validated.

The screen also identified the E2 ubiquitin conjugating enzyme UBE2G2 and the E3 ubiquitin ligase RNF145. Depletion of UBE2G2 prevented HMGCR degradation, implying that all ligases involved in HMGCR degradation utilise this E2 enzyme. In contrast, and despite being a high confidence hit in our screen, depletion of RNF145 caused a highly reproducible but small inhibition of sterol-accelerated HMGCR degradation, confirming the sensitivity of the screen to detect partial phenotypes and predicting the requirement for at least one additional UBE2G2-dependent ligase. A subsequent, targeted ubiquitome library screen in an RNF145-knockout reporter cell line confirmed a role for gp78 in HMGCR degradation. Gp78 has previously been shown to use UBE2G2 as its cognate E2 enzyme in the degradation of ERAD substrates (*Chen et al., 2006*). During preparation of this manuscript, the combined involvement of RNF145 and gp78 in Insig-mediated HMGCR degradation in hamster (CHO) cells was also reported (*Jiang et al., 2018*), confirming the role for these ligases in other species.

The availability of an RNF145-specific polyclonal antibody provides further insight into the expression and activity of endogenous RNF145, without the concerns of overexpression artefacts. RNF145 is an ER-resident E3 ubiquitin ligase with several unique features that make it well-suited for HMGCR regulation. A challenge facing all proteins responsible for cholesterol regulation is that the target they monitor, cholesterol, resides entirely within membranes. Like HMGCR and SCAP, RNF145 contains a putative sterol-sensing domain in its transmembrane region (*Cook et al., 2017*), suggesting that sterols may facilitate RNF145's association with Insigs. In contrast to Jiang et al., 2018, who reported a constitutive, sterol-independent association between ectopically expressed RNF145 and Insig-1 or −2, we find that endogenous RNF145 interacts with endogenous Insig-1 in a sterol-dependent manner (*Figure 6C*), as reported for the interaction of SCAP and HMGCR with the Insig proteins (*Lee et al., 2007*). The binding of RNF145 to Insig-1 is HMGCR-independent (*Figure 6D*). Furthermore, in the absence of Insigs, the RNF145-HMGCR association is lost (*Figure 6E*), implying that the interaction between these two proteins is absolutely Insig-dependent. Therefore, sterols trigger the recruitment of RNF145 to HMGCR via Insigs, leading to HMGCR ubiquitination and

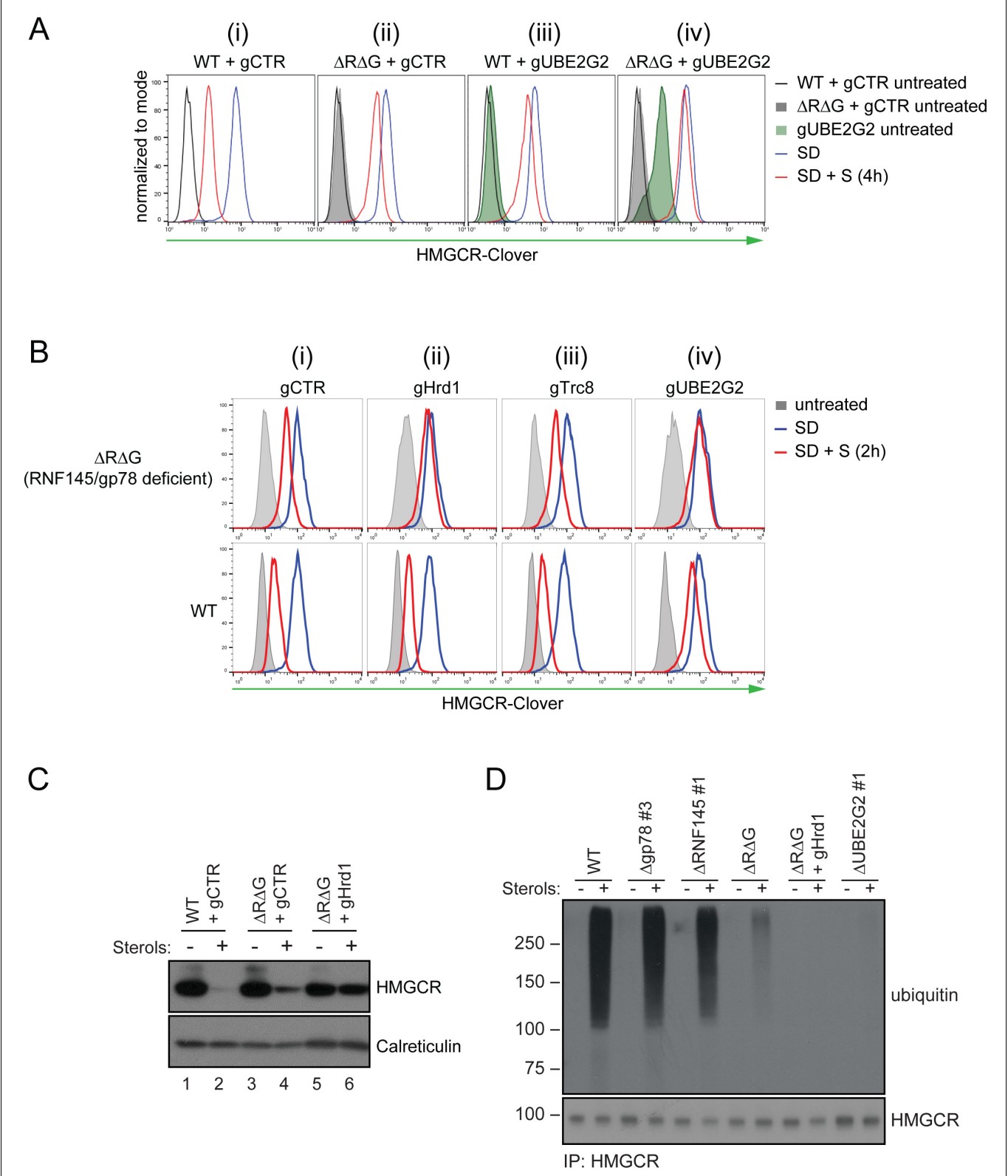

**Figure 7.** In the absence of RNF145 and gp78, Hrd1 targets HMGCR for ubiquitination and degradation. (**A**) Loss of gp78, RNF145 and UBE2G2 exert an additive effect on HMGCR degradation. WT or RNF145/gp78 double knockout (ΔRΔG #11) HMGCR-Clover HeLa cells transiently expressing sgRNAs targeting UBE2G2 (gUBE2G2) or B2M (gCTR) were enriched by puromycin selection, sterol-depleted (SD) overnight and HMGCR-Clover expression assessed ± sterols (S, 4 hr). Representative of 3 independent experiments. (**B and C**) A targeted gene approach shows that loss of Hrd1 from RNF145/

*Figure 7 continued on next page*

*Figure 7 continued*

gp78 double knockout cells blocks sterol-accelerated degradation of HMGCR. (B) WT and ΔRNF145 Δgp78 (ΔRΔG #11) HMGCR-Clover cells transfected with sgB2M (gCTR), a pool of 3-4 sgRNAs targeting either Hrd1 (gHrd1), TRC8 (gTRC8) or UBE2G2 (gUBE2G2), were sterol-depleted (SD, 20 hr) and HMGCR-Clover expression assessed by FACS analysis ± sterols (S, 2 hr). (C) WT and RNF145/gp78 double knockout cells (ΔRΔG #7) were transfected with a pool of four Hrd1-specific sgRNAs or gCTR, sterol-depleted overnight before addition of sterols (4 hr) and analysis by SDS-PAGE and immunoblotting. RNF145 and gp78 knockout validation is shown in *Figure 3—figure supplement 1B* (ΔRNF145 #1) and *Figure 3—figure supplement 3B* (ΔRΔG #7), respectively. (D) RNF145, gp78 and Hrd1 are required for sterol-accelerated HMGCR ubiquitination. HMGCR was immunoprecipitated (IP) from the indicated cell lines grown in sterol-depleted media (20 hr) ± sterols (1 hr). MG-132 (50 μM) was added 30 min before sterol supplementation. Ubiquitinated HMGCR was detected using an anti-ubiquitin antibody.

DOI: https://doi.org/10.7554/eLife.40009.022

The following figure supplements are available for figure 7:

**Figure supplement 1.** Combinatorial depletion of E3 ubiquitin ligases.

DOI: https://doi.org/10.7554/eLife.40009.023

**Figure supplement 2.** TRC8 depletion does not affect HMGCR-Clover degradation.

DOI: https://doi.org/10.7554/eLife.40009.024

degradation. This ability of RNF145 to rapidly bind Insigs following sterol availability supports a key role for this ligase in HMGCR regulation.

A striking feature of RNF145 is its short half-life and rapid proteasome-mediated degradation, which contrasts with the long-lived gp78 (*Figure 5A/B*, *Figure 5—figure supplement 1B*). RNF145 is an intrinsically unstable ligase whose half-life is regulated through autoubiquitination and was not prolonged on binding to Insig proteins (data not shown). Its stability and turnover is RING- and UBE2G2-dependent, but independent of either the gp78 (*Figure 6A- C* ) or Hrd1 E3 ligase (*Figure 5—figure supplement 1C*). As cells become sterol-depleted, the transcriptional increase in RNF145 (*Figure 5C/D*) likely anticipates the need to rapidly eliminate HMGCR, once normal cellular sterol levels are restored.

While this sterol-dependent transcriptional increase in RNF145 expression may at first seem counterintuitive, under sterol-deplete conditions the RNF145 ligase is not engaged with Insigs or its HMGCR substrate. The build-up of RNF145 predicts the restoration of sterol concentrations allowing RNF145 to immediately engage with, and degrade its HMGCR substrate. Thus the build-up of RNF145 anticipates its critical role in the restoration of cellular cholesterol homeostasis.

RNF145 transcription was reported to be regulated by the sterol-responsive Liver X Receptor (LXR) family of transcription factors (*Cook et al., 2017*; *Zhang et al., 2017*), which transcriptionally activate cholesterol efflux pumps (ABCA1, ABCG1) (*Costet, 2000*; *Edwards et al., 2002*) and the IDOL E3 ubiquitin ligase, which targets the LDLR for degradation (*Zelcer et al., 2009*). Pharmacological treatment of HeLa cells with the LXR inducer (GW3965) increased protein levels of ABCA1, but RNF145 transcript levels were not significantly increased (*Figure 5 – figure supplement 2A/B*). In HeLa cells, therefore, the increased expression of RNF145 following cholesterol starvation is not primarily driven by the LXR pathway.

While it is not unusual for more than one ligase to be required for substrate ERAD degradation (*Christianson and Ye, 2014*; *Morito et al., 2008*; *Stefanovic-Barrett et al., 2018*), the redundancy in HMGCR turnover is intriguing. This may simply reflect the central role of HMGCR in the mevalonate pathway and the importance of a fail-safe mechanism of HMGCR regulation to both maintain substrates for non-sterol isoprenoid synthesis and prevent cholesterol overproduction. Alternative explanations can also be considered, particularly as the properties of RNF145 and gp78 are so different. Under sterol-deplete conditions gp78 also regulates the degradation of Insig-1, but following addition of sterols, the association of Insigs with SCAP displaces Insigs binding to gp78 (*Yang et al., 2002*; *Lee et al., 2006*). Different Insig-associated complexes are therefore likely to co-exist within the ER membrane, under both sterol-replete and -deplete conditions, and will reflect the sterol microenvironment of the ER (*Goldstein et al., 2006*). Under these circumstances it might be advantageous to have more than one ligase regulating HMGCR. Alternatively, gp78 may provide basal control of the reductase, which can then be 'fine-tuned' by the sterol-responsive RNF145, reflecting the sterol concentration of the local ER environment. Further understanding of the stoichiometry and nature of the different Insig complexes within the ER membrane will be important. While all cells need to regulate their intracellular cholesterol, the contribution of each ligase to sterol regulation

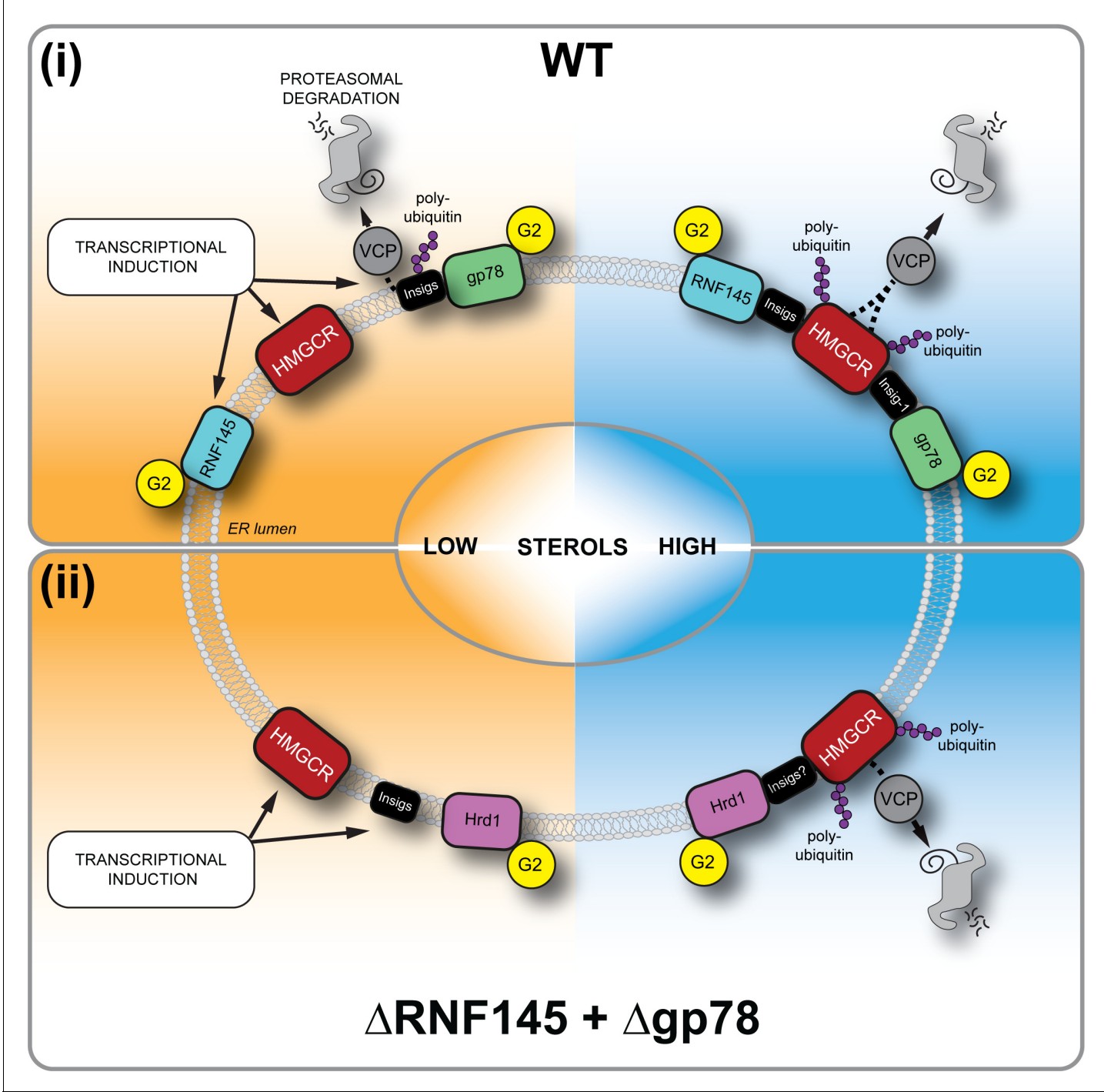

**Figure 8.** Sterol-induced HMGCR degradation by RNF145, gp78 and Hrd1. (i) Under sterol-depleted conditions (shaded orange), HMGCR, Insigs and RNF145 are transcriptionally induced leading to accumulation of RNF145 and HMGCR. Insigs are continually turned over by gp78-mediated polyubiquitination, extracted from the membrane by VCP and degraded by the 26S proteasome. HMGCR stability is dramatically increased as it is not engaged by either RNF145, gp78 or their shared E2 ubiquitin conjugating enzyme UBE2G2 (G2). In the presence of sterols (shaded blue), RNF145 and gp78 are recruited to HMGCR in an Insig-assisted fashion, mediating the sterol-accelerated and UBE2G2-dependent degradation of HMGCR by the ubiquitin-proteasome system. Under these conditions both RNF145 and gp78 can independently ubiquitinate HMGCR, which is then extracted from the ER membrane in a VCP-dependent manner. The stoichiometry and make-up of the different Insig complexes within the ER membrane are unknown. (ii) When both RNF145 and gp78 are not available (ΔRNF145 + Δgp78), Hrd1 and UBE2G2 can promote removal of HMGCR in the presence of sterols.
DOI: https://doi.org/10.7554/eLife.40009.025

may also depend on their differential tissue expression. In this regard, liver-specific ablation of gp78 in mice has been reported to lead to increased steady-state levels of hepatocyte HMGCR (*Liu et al., 2012*), whereas gp78 knockout MEFs show no apparent impairment in HMGCR degradation (*Tsai et al., 2012*). Further delineation of the contribution of each ligase to HMGCR degradation in different tissues and cell types will be important.

A role for the Hrd1 E3 ligase in HMGCR regulation was unanticipated, and both orthologues (gp78 and Hrd1) of yeast Hrd1p, which regulates yeast HMGCR (Hmg2p), are therefore involved in mammalian HMGCR turnover. The best recognised function of Hrd1 is the ubiquitination of mis-folded or unassembled ER-lumenal and membrane proteins targeted for ERAD (*Sato et al., 2009*; *Tyler et al., 2012*; *Christianson et al., 2008*). Our finding that Hrd1 is only involved in HMGCR reg-ulation when the other two ligases are absent, suggests that under sterol-rich conditions, and in the absence of RNF145 or gp78, conformational changes in the sterol-sensing domains of HMGCR may lead to a less ordered state and be recognised and targeted by the Hrd1 quality control pathway. Ligand-induced selective and reversible local misfolding in Hmg2p, dubbed 'mallostery', is a sug-gested mode of recognition by Hrd1p (*Wangeline et al., 2017*; *Wangeline and Hampton, 2018*).

The mechanism underlying recognition of HMGCR by Hrd1 is unclear, and whether the Hrd1 com-plex directly recognises sterol-induced structural changes as seen with Hmg2 degradation in yeast is unknown. Hrd1 might utilise the Insig proteins as scaffolds for HMGCR binding. This is partially borne out in the complete rescue of HMGCR in Insig-1 and −2 depleted cells (*Figure 1F*). These mechanisms are not mutually exclusive and suggest that the contributions by different ligases may represent a regulated misfolding event as part of a ligand-mediated control of HMGCR stability. Fur-ther investigation is needed to clearly determine their contribution to HMGCR regulation.

In summary, our unbiased approach to identify proteins involved in sterol-regulated HMGCR deg-radation resolves the ambiguity of the responsible E3 ubiquitin ligases, and uncovers additional con-trol points in modulating the activity of this important enzyme in health and disease.

# Materials and methods

**Key resources table**

| Reagent type | Designation | Source | Identifiers | Additional information |
|---|---|---|---|---|
| Antibody | anti-gp78 (rabbit polyclonal) | ProteinTech, 16675–1-AP | RRID:AB_2226463 | WB (1:1000) |
| Antibody | anti-ubiquitin (mouse monoclonal) | Life Sensors, VU101 | RRID:AB_2716558 | WB (1:1000) |
| Antibody | anti-V5 tag (mouse monoclonal) | Abcam, ab27671 | RRID:AB_471093 | WB (1:1000) |
| Antibody | anti-Insig-1 (rabbit polyclonal) | Abcam, ab70784 | RRID:AB_1269181 | WB (1:1000) |
| Antibody | anti-Hrd1 (rabbit polyclonal) | Abgent, AP2184a | RRID:AB_2199838 | WB (1:5000) |
| Antibody | anti-TRC8 (rabbit polyclonal) | Santa Cruz, sc-68373 | RRID:AB_2238721 | WB (1:2000) |
| Antibody | anti-HMGCR (mouse monoclonal) | Santa Cruz, sc-271595 | RRID:AB_10650274 | WB (1:1000) |
| Antibody | anti-UBE2G2 (mouse monoclonal) | Santa Cruz Biotechnology, sc-100613 | RRID:AB_1130984 | WB (1:1000) |
| Recombinant DNA reagent | pSpCas9(BB)—2A-Puro V1 (plasmid) | Addgene #48139 | n/a | |
| Recombinant DNA reagent | pSpCas9(BB)—2A-Puro V2 (plasmid) | Addgene #62988 | n/a | |
| Recombinant DNA reagent | pKLV-U6gRNA(BbsI)-PGKpuro2ABFP (plasmid) | Addgene # 50946 | n/a | |

*Continued on next page*

*Continued*

| Reagent type | Designation | Source | Identifiers | Additional information |
|---|---|---|---|---|
| Recombinant DNA reagent | genome-wide sgRNA library | other | n/a | kind gift from the Bassik lab (Stanford University), PMID: 28474669 |
| Recombinant DNA reagent | ubiquitome sgRNA library | this study | n/a | Generated by Lehner and Nathan labs |
| Peptide, recombinant protein | V5 peptide | Sigma-Aldrich | V7754-4MG | |
| Chemical compound, drug | lipoprotein-deficient serum (LPDS) | Biosera | FB-1001L/100 | |
| Chemical compound, drug | digitonin | Merck | 300410–5 GM | |
| Chemical compound, drug | mevastatin | Sigma-Aldrich | M2537-5MG | |
| Chemical compound, drug | mevalonolactone | Sigma-Aldrich | M4467-1G | |
| Chemical compound, drug | cholesterol | Sigma-Aldrich | C3045-5G | |
| Chemical compound, drug | 25-hydroxycholesterol | Sigma-Aldrich | H1015-10MG | |
| Chemical compound, drug | methyl-β-cyclodextrin (MBCD) | Sigma-Aldrich | 332615–1G | |
| Chemical compound, drug | NMS-873 | Selleckchem | s728501 | |
| Chemical compound, drug | cycloheximide | Sigma-Aldrich | C-7698 | |
| Chemical compound, drug | IgG Sepharose 6 Fast Flow | GE Healthcare | 17-0969-01 | |
| Chemical compound, drug | Protein A-Sepharose | Sigma-Aldrich | P3391-1.5G | |
| Chemical compound, drug | iodoacetamide | Sigma-Aldrich | I1149-5G | |
| Chemical compound, drug | cOmplete protease inhibitor | Roche | 27368400 | |
| Chemical compound, drug | phenylmethyl sulfonyl fluoride (PMSF) | Roche | 20039220 | |
| Chemical compound, drug | N-ethylmaleimide (NEM) | Sigma-Aldrich | E3876-5G | |
| Chemical compound, drug | puromycin | Cayman Chemicals | 13884 | |
| Chemical compound, drug | hygromycin B | Invitrogen | 10687010 | |

## Plasmids and expression constructs

Single guide RNAs (sgRNAs) were cloned into pSpCas9(BB)−2A-Puro V1 (Addgene #48139, deposited by Dr. Feng Zhang), pSpCas9(BB)−2A-Puro V2 (Addgene #62988, deposited by Dr. Feng Zhang) as previously described (*Ran et al., 2013*). The genome-wide sgRNA library (*Morgens et al., 2017*) was a kind gift from the Bassik lab (Stanford University). To generate the ubiquitome sgRNA library, sgRNAs (sgRNA sequences in *Supplementary file 1*) were cloned into pKLV-U6gRNA(BbsI)-PGKpuro2ABFP (Addgene # 50946) as reported previously (*Doench et al., 2016*). To generate RNF145 expression plasmids, the RNF145 CDS, PCR amplified from an RNF145 IMAGE clone (Source Bioscience, Nottingham, UK), was cloned into pHRSIN-P$_{SFFV}$-GFP-P$_{PGK}$-Hygromycin$^R$ (BamHI,

NotI) (*Demaison et al., 2002*), replacing GFP with the transgene. To create RNF145-V5, the RNF145 CDS was Gibson-cloned into pHRSIN-P$_{SFFV}$-P$_{PGK}$-Hygromycin$^R$ containing a downstream in-frame V5-tag. RNF145-V5 RING domain mutations (C552A, H554A) were introduced by PCR amplification of RNF145-V5 fragments with primers encoding C552A and H554A mutations and RNF145(C552A, H554A)-V5 was introduced into pHRSIN-P$_{SFFV}$-P$_{PGK}$-Hygromycin$^R$ by Gibson assembly. FLAG-NLS-Cas9 was cloned from the lentiCRISPR v2 (*Sanjana et al., 2014*) (Addgene #49535, deposited by Feng Zhang) into pHRSIN.pSFFV MCS(+) pSV40 Blast (BamHI, NotI).

## Compounds

The following compounds were used in this study: Dulbecco's Modified Eagle's Medium high glucose (DMEM; Sigma-Aldrich, 6429–500 ml), foetal calf serum (FCS; Seralab (catalogue no: EU-000, SLI batch: E8060012, Supplier batch: A5020012) and Life Technologies (catalogue no: 10270, lot: 42G4179K)), lipoprotein-deficient serum (LPDS; biosera, FB-1001L/100), mevastatin (Sigma-Aldrich, M2537-5MG), mevalonolactone (Sigma-Aldrich, M4467-1G), cholesterol (Sigma-Aldrich, C3045-5G), 25-hydroxycholesterol (Sigma-Aldrich, H1015-10MG), methyl-β-cyclodextrin (MBCD; Sigma-Aldrich, 332615–1G), GW3965 HCl (Sigma-Aldrich, G6295), bortezomib/PS-341 (BostonBiochem, I-200), (S)-MG132 (Cayman Chemicals, 10012628), NMS-873 (Selleckchem, s728501), digitonin (Merck, 300410–5 GM) (1% digitonin for immunoprecipitation experiments was generated by using the soluble supernatant of a 2% digitonin solution which had been left rotating with a small amount of CL4B beads overnight), cycloheximide (Sigma-Aldrich, C-7698), IgG Sepharose$^{TM}$ 6 Fast Flow (GE Healthcare, 17-0969-01), ProLong Gold Antifade Mountant with DAPI (Thermo Fisher), bovine serum albumin (BSA; Sigma-Aldrich, A4503-10G), Protein A-Sepharose$^R$ (Sigma-Aldrich, P3391-1.5G), iodoacetamide (IAA; Sigma-Aldrich, I1149-5G), cOmplete protease inhibitor (EDTA-free; Roche, 27368400), phenylmethylsulfonyl fluoride (PMSF; Roche, 20039220), V5 peptide (Sigma-Aldrich, V7754-4MG), N-ethylmaleimide (NEM; Sigma-Aldrich, E3876-5G), puromycin (Cayman Chemicals, 13884), hygromycin B (Invitrogen, 10687010), Penicillin-Streptomycin (10,000 U/mL; Thermo Fisher, 15140122).

## Antibodies

Antibodies specific for the following targets were used for immunoblotting analysis: Insig-1 (rabbit; Abcam, ab70784), Hrd1 (rabbit; Abgent, AP2184a), TRC8 (rabbit; Santa Cruz, sc-68373), tubulin (mouse; Sigma, T9026), VCP (mouse; abcam, ab11433), β-actin (mouse; Sigma-Aldrich, A5316), calnexin (mouse; AF8, kind gift from M Brenner, Harvard Medical School), calreticulin (rabbit; Pierce, PA3-900), HMGCR (mouse; Santa Cruz, sc-271595), HMGCR (rabbit; Abcam, ab174830), gp78 (rabbit; ProteinTech, 16675–1-AP), Insig-1 (rabbit; Abcam, ab70784), RNF145 (rabbit; ProteinTech, 24524-I-AP), V5 (mouse; Abcam, ab27671), VU-1 ubiquitin (mouse; Life Sensors, VU101), UBE2G2 (mouse; Santa Cruz, sc-100613), GFP (rabbit; Thermo Fisher Scientific, A11122), KDEL (mouse; Enzo, 10C3), HRP-conjugated anti-mouse and anti-rabbit (goat; Jackson ImmunoResearch), TrueBlot Anti-Rabbit-HRP (Rockland, 18-8816-31), TrueBlot Anti-Mouse-HRP ULTRA (Rockland, 18-8817-30). Alexa Fluor 488 (goat anti-rabbit; Thermo Fisher), Alexa Fluor 568 (goat anti-mouse; Thermo Fisher) were used as secondary antibodies for immunofluorescence microscopy. Anti-MHC-I (W6/32; mouse) and Alexa Fluor 647 (rabbit anti-mouse; Thermo Fisher) were used for cytofluorometric analysis.

## Cell culture

HeLa, HEK-293T, and HepG2 cells were maintained in DMEM +10% FCS+penicillin/streptomycin (5% CO$_2$, 37°C). HeLa cells were obtained from ECACC. HEK-293T and HepG2 cells were obtained from ATCC. HeLa and HepG2 cells were authenticated by STR profiling (Eurofins Genomics). All cell lines tested mycoplasma negative (Lonza MycoAlert). Transfection of HeLa cells was performed using the TransIT-HeLa MONSTER kit (Mirus) according to the manufacturer's instructions. In sort, cells were seeded at low confluency in 12-well tissue culture plates and the next day transfection mix (1 µg DNA, 3 µl TransIT-HeLa reagent +2 µl MONSTER reagent in OptiMEM (Gibco)) was added. Alternatively, reverse transfection was performed by seeding 3.5*10$^5$ cells per well of a 12-well plate to the transfection mix on the day of transfection. For co-transfection of multiple sgRNA plasmids, equal amounts of each plasmid were added up to 1 µg.

## CRISPR/Cas9-mediated gene knockout

CRISPR/Cas9-mediated genomic editing was performed according to Ran et al. (Ran et al., 2013). For generation of knockout cell lines, cells were transfected with pSpCas9(BB)−2A-Puro (PX459) V1.0 or V2.0 (Addgene #48139, and #62988 respectively; deposited by Dr. Feng Zhang) containing a sgRNA specific for the targeted gene of interest. Guide RNA sequences are listed in *Supplementary file 2*. Cells were cultured for an additional 24 hr before selection with puromycin (2 µg/ml) at low confluency for 72 hr. The resulting mixed knockout populations were used to generate single-cell clones by limiting dilution or fluorescence-assisted single-cell sorting. A detailed list of single cell knockout clones used in this study can be found in *Supplementary file 3*. Gene disruption was validated by immunoblotting, immunoprecipitation and/or targeted genomic sequencing.

## CRISPR/Cas9-mediated gene knock-in

An HMGCR-Clover knock-in donor template was created by Gibson assembly of ~1 kb flanking homology arms, PCR-amplified from HeLa genomic DNA, and the NsiI and PciI digested backbone from pMAX-GFP (Amaxa) cloned into the loxP-Ub-Puro cassette from pDonor loxP Ub-Puro (a kind gift from Ron Kopito, Stanford University). Each arm was amplified using nested PCR. The 5' arm was amplified using 5'-GATGCAGCACAGAATGTTGGTAG-3' and 5'-CAATGCCCATGTTCCAG TTCAG-3', followed by 5'-CAATGCCCATGTTCCAGTTCAG-3' and 5'-CAGCTGCACCATGCCATCTA TAG-3'. The 3' arm was amplified using the following primer pairs: 5'-CCAAGGAGCTTGCACCAA-GAAG-3' and 5'-CTAAGGTCCCAGTCTTGCTTG-3'. The product served as template for a subsequent PCR step using the primers 5'-CCAAGGAGCTTGCACCAAGAAG-3' and 5'-GTCACCCTCATC TAAGCAAC-3'. Overhangs required for Gibson assembly were introduced by PCR. HeLa cells were co-transfected with Cas9, sgRNA targeting immediately downstream of the HMGCR stop codon and donor template. Three different donor templates were simultaneously transfected, each differing in the drug resistance marker (puromycin, hygromycin and blasticidin). The transfected cells were treated with the three antibiotics five days *post* transfection until only drug-resistant cells remained. The resulting population was transfected with Cre-recombinase in pHRSIN MCS(+) IRES mCherry pGK Hygro. mCherry positive cells were single-cell cloned by FACS.

## Genetic validation of HMGCR-Clover knock-in cells

To confirm the knock-in of a myc- and Clover-tag downstream of the HMGCR coding sequence, genomic DNA was isolated from HeLa HMGCR-Clover cells using the Quick-gDNA MicroPrep kit (Zymo Research). The genomic sequence encoding the myc- and Clover-tags and flanking 5' and 3' homology regions were amplified using the following primer combination: 5'- ACTATTCATCTACTG TAGTTCCAAGTTAAAATTCTACACTC-3', 5'- GCATGTAAAGCACTAAACTGTGTTCAGATCTGAG-GAGTC-3'. PCR products were separated by agarose gel electrophoresis, gel excised and analysed by Sanger sequencing.

## Lentivirus production and transductions

HEK-293T cells were transfected with a lentiviral expression vector, the packaging vectors pCMVΔR8.91 and pMD.G at a ratio of 1:0.7:0.3 using TransIT-293 (Mirus) as recommended by the manufacturer. For production of CRISPR library virus, HEK-293T cells were transfected as above in 15 cm tissue culture plates. 48 hr *post* transfection, virus-containing media was collected, filtered (0.45 µm pore size) and directly added to target cells or frozen (−80°C) for long-term storage. Typically, cells were transduced in 6-well tissue culture plates at an M.O.I. <1 and selected with puromycin (2 µg/ml) or hygromycin B (200 µg/ml). To generate HeLa HMGCR-Clover stably expressing Cas9, HeLa HMGCR-Clover cells were transduced with pHRSIN-$P_{SFFV}$-Cas9-$P_{PGK}$-Hygromycin[R] and stable integrants selected with hygromycin B. Cas9 activity was confirmed by transduction with pKLV encoding a β−2-microglobulin (B2M)-targeting sgRNA followed by puromycin selection. MHC-I surface expression was assessed by flow cytometry in puromycin-resistant cells five days post transduction. Typically, ~90% reduction of cell surface MHC-I expression was observed.

## Fluorescent PCR

To identify CRISPR-induced frame-shift mutations, genomic DNA was extracted from wild type HeLa cells and RNF145 CRISPR clones using the Quick-gDNA MicroPrep kit (Zymo Research) followed by

nested PCR of the genomic region 5' and 3' of the predicted sgRNA binding site. One in each primer pair for the second PCR was 5' modified with 6-FAM™ (fluorescein, Sigma-Aldrich). Primer sequences were as follows: For sgRNA #8 PCR1_Forward: CAGAATGCTCACTAGAAGATTAG, PCR1_Reverse: GTAGTATACGTTCTCACATAG, PCR2_Forward: GTGATGTAGACACTCACCTAC and PCR2_Reverse: GTGACAACCTATTAGATTCGTG. PCR products were detected using an ABI 3730xl DNA Analyser.

## Flow cytometry and Fluorescence-activated cell sorting (FACS)

Cells were collected by trypsinisation and analysed using a FACS Calibur (BD) or an LSR Fortessa (BD). Flow cytometry data was analysed using the FlowJo software package. Cells resuspended in sorting buffer (PBS + 10 mM HEPES +2% FCS) were filtered through a 50 µm filter, and sorted on an Influx machine (BD), or, for the ubiquitome CRISPR/Cas9 screen, on a FACS Melody (BD). Sorted cells were collected in DMEM +50% FCS and subsequently cultured in DMEM +10% FCS+penicillin/ streptomycin. For MHC-I flow cytometric analysis, cells resuspended in cold PBS were incubated with W6/32 (20 min, 4°C), washed twice and then incubated with Alexa-647-labelled anti-mouse anti-body (15 min, 4°C). Cells were washed twice and resuspended in PBS.

## CRISPR/Cas9 knockout screens

For genome-wide and ubiquitome-library CRISPR/Cas9 knockout screens, $10^8$ and $1.2*10^7$ HeLa HMGCR-Clover (Cas9) or $\Delta$RNF145 #6 (Cas9), respectively, were transduced at M.O.I. ~ 0.3 by spin-fection (750xg, 60 min, 37°C). Transduction efficiency was determined *via* flow-cytometry-based measurement of mCherry (genome-wide screen) or BFP (ubiquitome screen) expression 48–72 hr *post* infection. Transduced cells were enriched by puromycin selection (2 µg/ml (genome-wide screen), 1 µg/ml (ubiquitome-library screen)). On day 8 (genome-wide screen) or day 7 (ubiquitome-library screen) post transduction, cells were rinsed extensively with PBS and cultured overnight in starvation medium (DMEM +10% LPDS+10 µM mevastatin +penicillin/streptomycin) before sterol addition (2 µg/ml 25-hydroxycholesterol and 20 µg/ml cholesterol for 5 hr). An initial FACS selection ('sort #1') on cells expressing high levels (~0.3–0.6% of overall population) of HMGCR-Clover (HMGCR-Clover$^{high}$) was performed. $2*10^5$ (genome-wide screen) and ~$10^5$ (ubiquitome-library screen) sorted cells were pelleted and DNA was extracted using the Quick-gDNA MicroPrep kit (Zymo Research). To gauge sgRNA enrichment, DNA was extracted from $3*10^6$ (genome-wide library screen) or $6*10^6$ (ubiquitome library screen) cells pre-sort using the Gentra Puregene Core kit A (Qiagen). Cells in the genome-wide screen were subjected to a second round of sterol deprivation and sort (see above) after expansion of initially $2.5*10^5$ sorted cells for 8 days. Sorted cells were cultured until $5*10^6$ cells could be harvested for genomic DNA extraction using the Gentra Puregene Core kit A (Qiagen). Individual integrated sgRNA sequences were amplified by two sequential rounds of PCR, the latter introducing adaptors for Illumina sequencing (*Supplementary file 4*). Sequencing was carried out using the Illumina HighSeq (genome-wide screen) and MiniSeq (ubiquitome-library screen) platforms. Illumina HiSeq data was analysed as described previously (*Timms et al., 2016*). Guide RNA counts were analysed with the RSA algorithm under default settings (*König et al., 2007*). Of note, a gene's calculated high significance value and therefore high enrichment in the selected population does not necessarily reflect its importance relative to genes with lower significance values/enrichment, since gene disruption can be incomplete or lethal phenotypes might evade enrichment.

## Quantitative PCR

Whole-cell RNA was isolated with the RNeasy Plus Mini Kit (Qiagen, Venlo, Netherlands) and reverse transcribed using Oligo(dT)15 primer (Promega, C110A) and SuperScript™ III reverse transcriptase (Invitrogen). Transcript levels were determined in triplicate using SYBR Green PCR Master Mix (Applied Biosystems) in a real time PCR thermocycler (7500 Real Time PCR System, Applied Biosystems). Primers used for target amplification can be found in *Supplementary file 5*. RNA quantification was performed using the $\Delta\Delta$CT method. GAPDH transcript levels were used for normalization. Raw data can be found in *Figure 5—source data 1* and *Figure 5—figure supplement 2—source data 1*.

## Sterol depletion assays

Typically, HeLa cells at ~50% confluency were washed five times with PBS and cultured for 16–20 hr in starvation medium (DMEM +10% LPDS+10 µM mevastatin +50 µM mevalonate +penicillin/strep-tomycin) before addition of 25-hydroxycholesterol (2 µg/ml) and cholesterol (20 µg/ml) to analyse sterol-accelerated protein degradation.

## Chol:MBCD complex preparation

Complexation of cholesterol (2.5 mM) with MBCD (25 mM) was performed according to Christian et al. (Christian et al., 1997). An emulsion of cholesterol powder (final: 2.5 mM) and an MBCD solution (25 mM) was produced by vortexing and tip sonication (1 min in 10 s intervals), and continuously mixed for 16 hr at 37°C. The solution was sterile filtered (0.45 µm PVDF pore size) and stored at −20°C.

## Preparation of sterols and mevalonate

Sterols were prepared by resuspension in ethanol or complexation with MBCD (see above). Mevalonate was prepared by adding 385 µl 2.04 M KOH to 100 mg mevalonolactone (Sigma). The solution was heated (1 hr, 37°C) and adjusted to a 50 mM stock solution.

## SDS-PAGE and immunoblotting

Cells were collected mechanically in cold PBS or by trypsinisation, centrifuged (1000xg, 4 min, 4°C), and cell pellets resuspended in lysis buffer (1% (w/v) digitonin, 1x cOmplete protease inhibitor, 0.5 mM PMSF, 10 mM IAA, 2 mM NEM, 10 mM TRIS, 150 mM NaCl, pH 7.4). After 40 min incubation on ice, lysates were centrifuged (17.000xg, 15 min, 4°C), the post-nuclear fraction isolated and protein concentration determined by Bradford assay. Samples were adjusted with lysis buffer and 6 x Laemmli buffer +100 mM dithiothreitol (DTT) and heated at 50°C (15 min). Samples were separated by SDS-PAGE and transferred to PVDF membranes (Merck) for immunodetection. Membranes were blocked in 5% milk +PBST (PBS + 0.2% (v/v) Tween-20) (1 hr) and incubated with primary antibody in PBST +2% (w/v) BSA at 4°C overnight. For detection from whole-cell lysate, membranes were incubated in peroxidase (HRP)-conjugated secondary antibodies. For detection of immunoprecipitated proteins, TrueBlot HRP-conjugated secondary antibodies (Rockland) were used. Immunoprecipitated RNF145 was detected using Protein A-conjugated HRP.

## Immunoprecipitation

Cells were seeded to 15 cm tissue culture plates ($4*10^6$ cells per plate). The following day, cells were washed five times with PBS and cultured in starvation medium (DMEM +10% LPDS+10 µM mevasta-tin +50 µM mevalonate +penicillin/streptomycin) for 20 hr. To prevent HMGCR membrane extraction and degradation, starved cells were treated with NMS-873 (50 µM) 0.5 hr prior to sterol addition (2 µg/ml 25-hydroxycholesterol and 20 µg/ml cholesterol for 1 hr) and collection in cold PBS. Cells were lysed in IP buffer 1 (1% (w/v) digitonin, 10 µM $ZnCl_2$, 1x cOmplete protease inhibitor, 0.5 mM PMSF, 10 mM IAA, 2 mM NEM, 10 mM TRIS, 150 mM NaCl, ph 7.4), post-nuclear fractions isolated by centrifugation (17.000xg, 4°C, 15 min) adjusted to 0.5% (w/v) digitonin and pre-cleared with IgG Sepharose™ 6 Fast Flow (1 hr). Endogenous RNF145 and V5-tagged RNF145 were immunoprecipitated at 4°C overnight from 3 to 6 mg whole-cell lysate using Protein A-Sepharose and anti-RNF145 or V5 antibody, respectively. Beads were collected by centrifugation (1500xg, 4 min, 4°C), washed for 5 min with IP buffer 2 (0.5% (w/v) digitonin, 10 µM $ZnCl_2$, 10 mM Tris, 150 mM NaCl, pH 7.4) and 4 × 5 min with IP buffer 3 (0.1% (w/v) digitonin, 10 µM $ZnCl_2$, 10 mM TRIS, 150 mM NaCl, pH 7.4). Proteins whose interaction with RNF145 was labile in the presence of 1% (v/v) Triton X-100 were recovered by eluting twice for 30 min with 20 µl TX100 elution buffer (1% (v/v) Triton X-100 +2 x cOmplete protease inhibitor in 10 mM TRIS, 150 mM NaCl pH 7.4) at 37°C under constant agitation. Immunoprecipitated RNF145 was subsequently eluted in 30 µl 2x Laemmli buffer +3% (w/v) DTT at 50°C (15 min). RNF145-V5 and associated complexes were recovered by two sequential elutions with V5 elution buffer (1 mg/ml V5 peptide +2 x cOmplete protease inhibitor in 10 mM TRIS, 150 mM NaCl pH 7.4) for 30 min at 37°C under continuous agitation. Eluted samples were adjusted with Laemmli buffer and denatured at 50°C (15 min).

## Ubiquitination assays

Cells were sterol-depleted (20 hr), treated with 20 µM MG132 and left for 30 min before addition of sterols (2 µg/ml 25-hydroxycholesterol and 20 µg/ml cholesterol for 1 hr) or EtOH (vehicle control). Immunoprecipitation of ubiquitinated HMGCR was performed as described above from 1 mg whole-cell lysate and using rabbit α-HMGCR (Abcam, ab174830). Proteins were eluted in 30 µl 2x Laemmli buffer +100 mM DTT at 50°C (15 min). For immunoblotting of ubiquitin with mouse VU-1 α-ubiquitin (Life Sensors, VU101), the PVDF membrane was incubated with 0.5% (v/v) glutaraldehyde/PBS pH 7.0 (20 min) and washed 3x with PBS prior to blocking in 5% (w/v) milk +PBS + 0.1% (v/v) Tween-20.

## Indirect immunofluorescence confocal microscopy

Cells were grown on coverslips, fixed in 4% PFA (15 min), permeabilised in 0.2% (v/v) Triton X-100 (5 min) and blocked with 3% (w/v) BSA/PBS (30 min). Cells were stained with primary antibody diluted in 3% (w/v) BSA/PBS (1 hr), washed with 0.1% (w/v) BSA/PBS, followed by staining with secondary antibody in 3% BSA/PBS (1 hr), an additional washing step (0.1% (w/v) BSA/PBS) and embedded using ProLong Gold Antifade Mountant with DAPI (Thermo Fisher). Images were acquired using an LSM880 confocal microscope (Zeiss) at 64x magnification.

## Statistical analysis

Statistical significance was calculated using the unpaired Student's t-test.

## Data deposition

Sequencing data from CRISPR/Cas9 knockout screens presented in this study have been deposited at the Sequence Read Archive (SRA) (genome-wide screen: SRP151225; ubiquitome screen: SRP151107).

## Acknowledgements

We are grateful to the following for their help in this study: Michael Bassik (Stanford University) kindly shared the genome-wide CRISPR/Cas9 sgRNA library. Ron Kopito (Stanford University) kindly donated the pDonor loxP Ub-Puro plasmid. FACS experiments were enabled by R Schulte and his FACS core facility team in CIMR. Stuart Bloor (CIMR), Gordon Dougan, Richard Rance and Nathalie Smerdon (Sanger Institute) assisted with Illumina sequencing. This work was supported by the Wellcome Trust, through a Principal Research Fellowship to PJL (210688/Z/18/Z), a Wellcome Trust Senior Clinical Research Fellowship to JAN (102770/Z/13/Z) and a Welcome Trust PhD studentship to SM. The CIMR is in receipt of a Wellcome Trust strategic award (100140).

## Additional information

### Funding

| Funder | Grant reference number | Author |
| --- | --- | --- |
| Wellcome Trust | 210688/Z/18/Z | Paul J Lehner |
| Wellcome Trust | 102770/Z/13/Z | James A Nathan |
| Wellcome Trust | PhD studentship | Sam A Menzies |

The funders had no role in study design, data collection and interpretation, or the decision to submit the work for publication.

### Author contributions

Sam A Menzies, Norbert Volkmar, Conceptualization, Data curation, Software, Formal analysis, Validation, Investigation, Visualization, Methodology, Writing—original draft, Project administration, Writing—review and editing; Dick JH van den Boomen, Conceptualization, Data curation, Formal analysis, Investigation, Methodology; Richard T Timms, Conceptualization, Writing—review and editing; Anna S Dickson, James A Nathan, Resources, Methodology, Writing—review and editing; Paul J

Lehner, Conceptualization, Supervision, Funding acquisition, Writing—original draft, Project administration, Writing—review and editing

Author ORCIDs
Norbert Volkmar (iD) http://orcid.org/0000-0003-0766-5606
James A Nathan (iD) http://orcid.org/0000-0002-0248-1632
Paul J Lehner (iD) http://orcid.org/0000-0001-9383-1054

Decision letter and Author response
Decision letter https://doi.org/10.7554/eLife.40009.037
Author response https://doi.org/10.7554/eLife.40009.038

---

## Additional files

Supplementary files
• Supplementary file 1. sgRNA sequences and genes targeted by the CRISPR/Cas9 ubiquitome library.
DOI: https://doi.org/10.7554/eLife.40009.026

• Supplementary file 2. sgRNA sequences for generation of knockout cell lines.
DOI: https://doi.org/10.7554/eLife.40009.027

• Supplementary file 3. Genetically modified cell lines used in this study.
DOI: https://doi.org/10.7554/eLife.40009.028

• Supplementary file 4. Primers used in CRISPR/Cas9 screens.
DOI: https://doi.org/10.7554/eLife.40009.029

• Supplementary file 5. Primer sequences used for qPCR.
DOI: https://doi.org/10.7554/eLife.40009.030

• Transparent reporting form
DOI: https://doi.org/10.7554/eLife.40009.031

Data availability
Sequencing data from CRISPR/Cas9 knockout screens presented in this study have been deposited at the Sequence Read Archive (SRA) (genome-wide screen: SRP151225; ubiquitome screen: SRP151107).

The following datasets were generated:

| Author(s) | Year | Dataset title | Dataset URL | Database and Identifier |
|---|---|---|---|---|
| Sam A. Menzies, Norbert Volkmar, Dick J. van den Boomen, Richard T. Timms Anna S. Dickson, James A. Nathan and Paul J. Lehner | 2018 | Genome-wide CRISPR screen in HeLa HMGCR-Clover cells | https://www.ncbi.nlm.nih.gov/sra/SRP151225 | Sequence Read Archive, SRP151225 |
| Sam A. Menzies, Norbert Volkmar, Dick J. van den Boomen, Richard T. Timms Anna S. Dickson, James A. Nathan and Paul J. Lehner | 2018 | Ubiquitome library screen in HeLa HMGCR-Clover RNF145 KO cells | https://www.ncbi.nlm.nih.gov/sra/SRP151107 | Sequence Read Archive, SRP151107 |

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
