## [Decision Letter]

[**Editorial note:** This article has been through an editorial process in which the authors decide how to respond to the issues raised during peer review. The Reviewing Editor's assessment is that all the issues have been addressed.]

Thank you for submitting your article "The sterol-responsive RNF145 E3 ubiquitin ligase mediates degradation of HMG-CoA reductase together with gp78 and Hrd1" for consideration by *eLife*. Your article has been reviewed by three peer reviewers, including Wade Harper as the Reviewing Editor and Reviewer #1, and the evaluation has been overseen by Ivan Dikic as the Senior Editor. The following individual involved in review of your submission has agreed to reveal their identity: Randy Y Hampton (Reviewer #2). A further reviewer remains anonymous.

The Reviewing Editor has highlighted the concerns that require revision and/or responses, and we have included the separate reviews below for your consideration. If you have any questions, please do not hesitate to contact us.

Summary:

This paper employs a clever assay to perform a genome-wide CRISPR screen for genes that control sterol-dependent turnover of HMGCR in the ER membrane. This is an exciting paper that uncovers a clear role for RNF145 in turnover of HMGCR, with additional roles for gp78 and Hrd1. The identity to E3s regulating HMGCR turnover in mammals has been controversial, with some evidence for gp78 and TRC8 in its turnover, and some evidence against these proteins being involved. Given the important of the pathway in controlling lipid regulatory mechanisms, understanding the key regulatory mechanisms is important. A second interesting aspect of the paper is the autoregulation of RNF145 itself, probably through an auto-ubiquitylation mechanism, although the data need further clarification as described below. Overall, the biochemical and screening data is of very high quality and convincing.

The reviewers were all quite positive about the paper.

Major concerns:

Most of the comments raised had to do with aspects of the text. Several suggestions were made on ways to improve clarity and to explain particular aspects of what can be concluded from the results or what prior work might say. In particular, the sterol regulation of RNF145 is a bit counterintuitive, as indicated by both reviewers 2 and 3. So we would strongly suggest that this be addressed, at a minimum through changes in the text, and if possible through an experiment (see reviewer 3 comments). The reviewers also suggest that you discuss the previous paper on RNF145 in greater detail. We also feel that addressing point 5 by reviewer 3 would improve the paper, in the event that this data is available or could be obtained quickly.

Separate reviews (please respond to each point):

*Reviewer #1:*

This paper reports a crispr screen leading to the identification of genes that control sterol-dependent turnover of HMGCR. The authors set up a cell line that reports on the abundance of a fluorescently tagged HMGCR protein (Figure 1) and performed a crispr screen (Figure 2). The identity to E3s regulating HMGCR turnover in mammals has been controversial, with some evidence for gp78 and TRC8 in its turnover, and some evidence against these proteins being involved. Given the important of the pathway in controlling lipid regulatory mechanisms, understanding the key regulatory mechanisms is important.

Through the genome wide screen, the authors identified Ube2g2 and RNF145 as regulators of HMGCR turnover. The show that gp78 and RNF145 are functionally redundant, and are backed up to a small degree by the Hrd1 E3. RNF145 itself has a relatively subtle phenotype and a secondary ubiquitylome screen in RNF145 null cells led to the identification of gp78 as the partially redundant function.

The authors have also found that RNF145 may be auto-regulated and is rapidly turned over under steady state conditions in a manner that requires its RING domain. They also found that its transcription and abundance is regulated by sterols and that the interaction of RNF145 with insig is sterol dependent, which are interesting findings and provide novel mechanistic insight.

Overall the paper is clear and to a large degree straightforward. The screening design is good. The results clarify the role for various E3s in HMGCR turnover, identify a new component, and deduce several regulatory aspects of the pathway. Overall, the paper is appropriate for *ELife*.

Minor:

1) The deltaR/deltaG nomenclature is a bit confusing and it might be better to simply write RNF145-/-;gp78-/-.

Additional data files and statistical comments:

All the CRISPR screening data (Sequence reads) should be provided.

*Reviewer #2:*

The paper by Menzies et al. out of the Lehrner lab is a typical example of the interesting and beautiful work that is routinely produced by this group. The experimental approach is clear and elegant (and I have never even heard of that fluorescent protein!). The data are clean and very informative, and the results are surprising and take us…some…or maybe most…of the way to resolving the Great Mammalian HMGcR Mystery.

If I understand the instructions to the reviewers, it is the case is that we are simply to provide suggestions; that this is an example of a new experimental mode of reviewing that makes it entirely up to the authors to accept suggestions and to respond or not as they see fit. That is a great development in terms of lessening the stranglehold that reviewers have on the fates of authors, and I approve, since I, and probably everyone who is reading this be them author or reviewer, has had that situation, where 50-60 thousand dollars later, the "requested additional experiment" didn't work or did work but didn't really enhance the strength of the paper. So good for *eLife*!

On the other hand, shoot, we already aren't getting paid, now we don't really have any authoritative say either. I feel more like a spectator in the courtroom, or a viewer of a newsfeed than a bone fide reviewer. But not a problem! This work is sufficiently well crafted that my optional suggestions will be offered in the spirit of optimistic encouragement.

1) The TRC8 problem: so the one issue missing is the absence of any TRC8 dependence in these effects. I am comfortable with that; I think a lot of these differing results are due to subtilties of gene expression that change which ligases, or regulating factors, appear or do not in a given experiment. Any discussion-based explanations for this?

2) A physiological model for RNF145's regulation: It is odd, is it not, that depleted sterols induce the E3 that will get rid of HMGCR induced by depleted sterols. What do you think the function may be? It would be interesting to hear what you think.

3) RNF145 information: It might be useful to emphasize (I didn't see it in the paper but perhaps I missed it) that RNF145 has a large transmembrane domain with an SSD. The SSD is an incredibly interesting module in biology, and especially sterol biology, and it should at least be mentioned.

4) The misfolding/quality control connection to HMGCR regulation: One of the very lovely things about this work is that the authors, through estimable and intense hard work, unveiled a role for the eponymous E3 of HMGCR regulation, namely Hrd1, when enough ligases are taken out of the picture. This is very cool. Two things. First, it is worth mentioning the earlier Debose-Boyd work showing that when mammalian HMGCR is expressed in *Drosophila* cells with the corresponding mammalian INSIG, one can observed, amazingly enough, *Drosophila* Hrd1 dependent. sterol-regulated degradation of the exogenous HMGCR. In fact this is a feature of a "misfolding-based" explanation for the HMGCR ligase dilemma that is also nicely supported by the work of these new studies. The idea being that in the original HRD system, that is yeast Hmg2 regulation by Hrd1 (nicely referenced in this new work), regulated misfolding is at the heart of that version of HMGCR regulation. Since protein quality control often can be mediated by a number of ligases that share ability to recognize misfolding, in the case of the mammal, perhaps regulated misfolding also plays a role: maybe the ability of different groups to observed different ligase dependencies is due to a certain shared ability among distinct ligases to recognize a regulated misfolding event as part of ligand-mediated control of HMGCR stability. Thus, the higher flexibility of quality control based regulation might allow a number of distinct quality control ligases to participate depending on details of presence, levels, and cell type. It is only one explanation, but it is an interesting one, and one we described in some detail and would appreciate you referencing from our Annual Reviews of Cell and Developmental Biology (Wangeline, Vashistha and Hampton, 2017). In fact we even suggest that this would be one place where CRISPR based analysis would provide some tests and resolution of this idea. Of course, I am biased a little, since I am Hampton, one of the authors. But it would be resonant with your new work and my recently graduated, first author student would also really appreciate it too.

Anyway, beautiful work. Congratulations!

Additional data files and statistical comments:

Very rigorous.

*Reviewer #3:*

Starting from a clever CRISPR screen, the authors report that the ER E3 ligase RNF145 cooperates with gp78 to control the sterol-regulated degradation of HMGCR. In their absence, Hrd1 acts in this role. Together, these 3 ER E3 ligases account for HMGCR sterol-regulated degradation in HeLa cells. The data are clearly presented and support the conclusions. The findings are consistent with a JBC paper published in March 2018 showing a role for RNF145 in HMGCR degradation in CHO cells. These findings are clinically important since they pertain to control of HMGCR protein levels and statin therapy and will be of interest to a broad audience.

Major comments:

1) Authors conclude that the HMGCR-Clover gene is present at a single locus and that these cells express endogenous HMGCR from another allele. This is based on the presence of HMGCR that co-migrates with HMGCR in parent cells. However, this could be due to proteolytic cleavage between HMGCR and Clover. This conclusion should be validated by PCR-sequencing.

2) The authors should include in their discussion comments on whether other genes identified in screens are likely to be positive hits and why. For example, is GALNT11 a true positive? Why might they have gotten EHD1? Could this disrupt LDLR trafficking, which was also found?

3) Zelcer and colleagues reported that RNF145 is an LXR target gene and induced by LXR agonist (Cook et al., 2017). LXR is typically activated under conditions of cholesterol excess. However, the authors find the opposite. Rather RNF145 is induced under sterol depleted conditions. To address this discrepancy, suggest that authors test effects of LXR agonist in parallel to sterol depletion in their cells and discuss these results.

4) These studies were obviously performed in parallel to those described in Jiang et al., 2018. While the authors mention this paper, they should discuss their own results in the context of this other study.

5) Evidence for a RNF145-Insig-HMGCR complex is strong. However, the authors depict RNF145 binding to Insig directly and not HMGCR in Figure 8. Suggest that authors test whether the RNF145-HNGCR interaction requires Insig1/2 to support this model or cite the work of others as support for this specific depiction.

Minor comments:

1) Please describe in the results why the p97 inhibitor was included in Figure 6 experiments and whether addition is required for interaction.

2) Figure 8 is a beautiful figure. However, the arrows next to STEROLS in the center are somewhat confusing and appear to indicate that sterols are moving between the top and bottom panels. Suggest writing "Low sterol" and "High sterol" or something similar to improve figure clarity.

3) To improve clarity, please describe in more detail the biology behind the US2-MHC1 assay used to validate TRC8 knockout.

4) Introduction paragraph three is missing a word, "despite confirming a for gp78"

5) Results first paragraph subheading, typo HMMGCR.

---

## [Author Response]

Reviewer #1:

[…] Overall the paper is clear and to a large degree straightforward. The screening design is good. The results clarify the role for various E3s in HMGCR turnover, identify a new component, and deduce several regulatory aspects of the pathway. Overall, the paper is appropriate for ELife.Minor:1) The deltaR/deltaG nomenclature is a bit confusing and it might be better to simply write RNF145-/-;gp78-/-.

Thank you for this suggestion. We did try this out, but due to simplicity and spatial constraints we think that the designation ‘ΔRΔG’ is equally appropriate as long as it is clearly stated in the figure legends and text. We prefer to avoid using the denomination ‘-/-’, since it implies complete knockout in a diploid system, while in HeLa cells ploidy is altered.

Additional data files and statistical comments:All the CRISPR screening data (Sequence reads) should be provided.

Raw sequence reads have been deposited on the Sequence Read Archive (SRA). Deposition IDs are listed in the Materials and methods section “Data deposition”.

Reviewer #2:

The paper by Menzies et al. out of the Lehrner lab is a typical example of the interesting and beautiful work that is routinely produced by this group. […] This work is sufficiently well crafted that my optional suggestions will be offered in the spirit of optimistic encouragement.1) The TRC8 problem: so the one issue missing is the absence of any TRC8 dependence in these effects. I am comfortable with that; I think a lot of these differing results are due to subtilties of gene expression that change which ligases, or regulating factors, appear or do not in a given experiment. Any discussion-based explanations for this?

We are aware of the TRC8 controversy and have worked with this ligase for many years. Throughout this project, despite extensive testing, we have never seen any involvement of TRC8 in the degradation of HMGCR in HeLa cells – consistent with the findings of the Weissman group. Clearly, we cannot absolutely rule out that TRC8 might play a role in HMGCR degradation in other cell lines or under different conditions. However, we are confident that in all the cell lines that we have tested, where we see a striking role for TRC8 in the HCMV-US2-mediated degradation of MHC-I and a host of other immunoreceptors (Stagg et al., 2009; Hsu et al., 2015), we are unable to see any role for TRC8 in HMGCR degradation.

2) A physiological model for RNF145's regulation: It is odd, is it not, that depleted sterols induce the E3 that will get rid of HMGCR induced by depleted sterols. What do you think the function may be? It would be interesting to hear what you think.

Our finding that sterol depletion induces RNF145 levels while HMGCR remains stable may at first seem somewhat counterintuitive. However, RNF145 is not associated with either the Insigs or HMGCR under these conditions (Figure 6A-C). We would therefore argue that the build-up of RNF145 under sterol depletion anticipates and facilitates rapid removal of HMGCR, once homeostasis is restored. This point has now been included in the Discussion section:

“While this sterol-dependent transcriptional increase in RNF145 expression may at first seem counterintuitive, under sterol-deplete conditions the RNF145 ligase is not engaged with Insigs or its HMGCR substrate. The build-up of RNF145 predicts the restoration of sterol concentrations allowing RNF145 to immediately engage with, and degrade its HMGCR substrate. Thus the build-up of RNF145 anticipates its critical role in the restoration of cellular cholesterol homeostasis.”

3) RNF145 information: It might be useful to emphasize (I didn't see it in the paper but perhaps I missed it) that RNF145 has a large transmembrane domain with an SSD. The SSD is an incredibly interesting module in biology, and especially sterol biology, and it should at least be mentioned.

We agree that this is an incredibly interesting module and did mention the presence of a putative SSD in RNF145 in both the results and the Discussion section of the original submission. We were, and still are, excited by this module and have made a number of mutations in the SSD of RNF145. Unfortunately this so destabilised the RNF145 protein that it was impossible to obtain any meaningful results. We have emphasised the potential contribution of the SSD to RNF145 activity:

*“*Like HMGCR and SCAP, RNF145 contains a putative sterol-sensing domain in its transmembrane region (Cook et al., 2017), suggesting that sterols may facilitate RNF145’s association with Insigs. In contrast to Jiang et al., who reported a constitutive, sterol-independent association between ectopically expressed RNF145 and Insig-1 or -2, we find that endogenous RNF145 interacts with endogenous Insig-1 in a sterol-dependent manner (Figure 6C), as reported for the interaction of SCAP and HMGCR with the Insig proteins (Lee et al., 2007). The binding of RNF145 to Insig-1 is HMGCR-independent (Figure 6D). Furthermore, in the absence of Insigs, the RNF145-HMGCR association is lost (Figure 6E), implying that the interaction between these two proteins is absolutely Insig-dependent. Therefore, sterols trigger the recruitment of RNF145 to HMGCR via Insigs, leading to HMGCR ubiquitination and degradation. This ability of RNF145 to rapidly bind Insigs following sterol availability supports a key role for this ligase in HMGCR regulation.”

4) The misfolding/quality control connection to HMGCR regulation: One of the very lovely things about this work is that the authors, through estimable and intense hard work, unveiled a role for the eponymous E3 of HMGCR regulation, namely Hrd1, when enough ligases are taken out of the picture. This is very cool. Two things. First, it is worth mentioning the earlier Debose-Boyd work showing that when mammalian HMGCR is expressed in Drosophila cells with the corresponding mammalian INSIG, one can observed, amazingly enough, Drosophila Hrd1 dependent. sterol-regulated degradation of the exogenous HMGCR. In fact this is a feature of a "misfolding-based" explanation for the HMGCR ligase dilemma that is also nicely supported by the work of these new studies. The idea being that in the original HRD system, that is yeast Hmg2 regulation by Hrd1 (nicely referenced in this new work), regulated misfolding is at the heart of that version of HMGCR regulation. Since protein quality control often can be mediated by a number of ligases that share ability to recognize misfolding, in the case of the mammal, perhaps regulated misfolding also plays a role: maybe the ability of different groups to observed different ligase dependencies is due to a certain shared ability among distinct ligases to recognize a regulated misfolding event as part of ligand-mediated control of HMGCR stability. Thus, the higher flexibility of quality control based regulation might allow a number of distinct quality control ligases to participate depending on details of presence, levels, and cell type.

This is a very interesting hypothesis and, we agree, not necessarily mutually exclusive with Insig-mediated HMGCR degradation in mammals. We have adjusted our Discussion section to that effect and included additional references:

*“*Our finding that Hrd1 is only involved in HMGCR regulation when the other two ligases are absent, suggests that under sterol-rich conditions, and in the absence of RNF145 or gp78, conformational changes in the sterol-sensing domains of HMGCR may lead to a less ordered state and be recognised and targeted by the Hrd1 quality control pathway. Ligand-induced selective and reversible local misfolding in Hmg2p, dubbed “mallostery”, is a suggested mode of recognition by Hrd1p (Wangeline et al., 2017; Wangeline and Hampton, 2018).”

*“*The mechanism underlying recognition of HMGCR by Hrd1 is unclear, and whether the Hrd1 complex directly recognises sterol-induced structural changes as seen with Hmg2 degradation in yeast is unknown. Hrd1 might utilise the Insig proteins as scaffolds for HMGCR binding. This is partially borne out in the complete rescue of HMGCR in Insig-1 and -2 depleted cells (Figure 1F). These mechanisms are not mutually exclusive and suggest that the contributions by different ligases may represent a regulated misfolding event as part of a ligand-mediated control of HMGCR stability. Further investigation is needed to clearly determine their contribution to HMGCR regulation.”

It is only one explanation, but it is an interesting one, and one we described in some detail and would appreciate you referencing from our Annual Reviews of Cell and Developmental Biology (Wangeline, Vashistha and Hampton, 2017). In fact we even suggest that this would be one place where CRISPR based analysis would provide some tests and resolution of this idea. Of course, I am biased a little, since I am Hampton, one of the authors. But it would be resonant with your new work and my recently graduated, first author student would also really appreciate it too.Anyway, beautiful work. Congratulations!

Reviewer #3:

[…] These findings are clinically important since they pertain to control of HMGCR protein levels and statin therapy and will be of interest to a broad audience.Major comments:1) Authors conclude that the HMGCR-Clover gene is present at a single locus and that these cells express endogenous HMGCR from another allele. This is based on the presence of HMGCR that co-migrates with HMGCR in parent cells. However, this could be due to proteolytic cleavage between HMGCR and Clover. This conclusion should be validated by PCR-sequencing.

The reviewer is correct. To further validate the HMGCR-Clover cell line we PCR-amplified and sequenced the HMGCR knock-in locus, including flanking sequences, to distinguish between the wild type and endogenously tagged HMGCR. Results are now shown in a new supplemental figure (Figure 1—figure supplement 1). This confirms the presence of both modified and unmodified HMGCR. These data have been referenced in the Results section:

“Residual, untagged HMGCR detected by immunoblot analysis in the reporter cells under sterol-depleted conditions suggested that at least one HMGCR allele remained untagged (Figure 1C, compare lanes 2 and 5), which was confirmed by PCR-amplification and sequencing of the genomic locus (Figure 1—figure supplement 1A-C).”

A new figure legend for Figure 1—figure supplement 1 was added:

“Figure 1—figure supplement 1. Genotyping of HMGCR-Clover knock-in cells.

(A) Schematic representation of PCR amplification of the genomic locus targeted for Clover insertion in HMGCR-Clover knock-in (KI) cells. Primers (‘1’, ‘2’) specifically annealing in the 5’ and 3’ homology arms flanking the Clover gene sequence were used to amplify a region encompassing 347 bp (i) in the WT allele and 1122 bp (ii) if the Clover insert is present.

(B) The PCR reaction described in (A) was performed using genomic DNA derived from parental (WT) and HMGCR-Clover knock-in HeLa cells. Amplicons derived from the unmodified (WT, 347 bp) and HMGCR-Clover knock-in allele (KI, 1122 bp) are indicated. As described in Figure 1C, HeLa HMGCR-Clover KI cells retain at least one unmodified HMGCR allele in addition to endogenously Clover-tagged HMGCR.

(C) Genetic profiling of the HMGCR-Clover KI locus in HMGCR-Clover HeLa cells. The longer 1122 bp PCR product, obtained as described in (A), was isolated by agarose gel electrophoresis and confirmed by Sanger sequencing.”

2) The authors should include in their discussion comments on whether other genes identified in screens are likely to be positive hits and why. For example, is GALNT11 a true positive? Why might they have gotten EHD1? Could this disrupt LDLR trafficking, which was also found?

Thank you for raising this point. Genes identified in the genome-wide screen did include a number of unexpected hits. Several of these candidates were subsequently validated, therefore representing true-positives. A discussion of their involvement in HMGCR degradation has now been included in the revised manuscript:

“For a small number of validated hits from our screen (Figure 2C, Figure 2—figure supplement 1D), the effects on sterol-accelerated HMGCR degradation were unanticipated and likely reflect wide-ranging alterations to the protein and lipid environment. Trans-2,3-enoyl CoA reductase (TECR) catalyses the final steps in the synthesis of very long-chain fatty acids (VLCFAs) (Moon and Horton, 2003) as well as the saturation step in sphingolipid degradation (Wakashima et al., 2014). Polypeptide N-acetylgalctosaminyltransferase 11 (GALNT11) initiates protein O-linked glycosylation, suggesting that a protein involved in HMGCR regulation requires O-linked glycosylation (Schwientek et al., 2002). Interestingly, our screen revealed that loss of the LDLR impaired HMGCR-Clover degradation. This finding is unexpected as the cholesterol added to the cells to induce HMGCR degradation was not in the form of LDL. EH domain-containing protein 1 (EHD1) is required for the internalisation and recycling of several plasma membrane receptors, including the LDLR (Naslavsky et al., 2007; Naslavsky and Caplan, 2011) and loss of EDH1 impairs LDLR trafficking with decreased intracellular cholesterol levels (Naslavsky et al., 2007). The other significant hits in the screen (PPAP2C, FER) have not been validated.”

3) Zelcer and colleagues reported that RNF145 is an LXR target gene and induced by LXR agonist (Cook et al., 2017). LXR is typically activated under conditions of cholesterol excess. However, the authors find the opposite. Rather RNF145 is induced under sterol depleted conditions. To address this discrepancy, suggest that authors test effects of LXR agonist in parallel to sterol depletion in their cells and discuss these results.

We thank the reviewer and we are aware of this discrepancy. Data presented in our study (Figure 5C, D) show a clear transcriptional induction of RNF145 under conditions of sterol depletion, correlating with a comparable fold-increase of RNF145 protein levels. We agree that this finding suggests a different mechanism of RNF145 transcriptional control from the Zelcer lab study (Cook et al., 2017), where treatment with an LXR agonist induced a small increase in RNF145 transcription. Despite the claims in the title of this paper this increase (Figure 1B Cook et al., 2017) was only 1.2 – 1.8 fold. Since LXRs are usually active under conditions of cholesterol excess, this is at odds with our observations. To clarify this discrepancy, we tested the effect of LXR induction on endogenous RNF145 mRNA and protein levels under sterol-depleted or –replete conditions in our system. Our new findings are now shown in Figure 5—figure supplement 2 and summarised below.

Treatment of HeLa cells with the LXR inducer GW3965 (also used in Cook et al., 2017), did not affect RNF145 mRNA levels, in clear contrast to the *bona fide* LXR target gene ABCA1, whose gene product was markedly increased (Figure 5 —figure supplement 2A, compare lanes 2 and 4 with lanes 1 and 3). Similarly, we did not observe a significant potentiating effect of LXR induction on RNF145 transcript levels under sterol-depleted conditions (Figure 5—figure supplement 2B). Our results clearly demonstrate that sterol depletion – not excess – favours RNF145 transcription.

These new findings have been incorporated into the Discussion section:

“RNF145 transcription was reported to be regulated by the sterol-responsive Liver X Receptor (LXR) family of transcription factors (Cook et al., 2017; Zhang et al., 2017), which transcriptionally activate cholesterol efflux pumps (ABCA1, ABCG1) (Costet, 2000; Edwards et al., 2002) and the IDOL E3 ubiquitin ligase, which targets the LDLR for degradation (Zelcer et al., 2009). Pharmacological treatment of HeLa cells with the LXR inducer (GW3965) increased protein levels of ABCA1, but RNF145 transcript levels were not significantly increased (Figure 5—figure supplement 2A/B). In HeLa cells, therefore, the increased expression of RNF145 following cholesterol starvation is not primarily driven by the LXR pathway.”

A new figure legend for Figure 5—figure supplement 2 has been added:

“Figure 5—figure supplement 2. Increased RNF145 transcription upon sterol depletion is LXR-independent.

(A) Hela cells were sterol-depleted (SD, 10% LPDS + 10 μM mevastatin + 50 μM mevalonate) for 48 h in the presence of the synthetic LXR ligand GW3965 (1 μM) or DMSO (vehicle control). Whole–cell lysates were analysed by SDS-PAGE and immunoblot assay. Representative of ≥ 3 independent experiments.

(B) HeLa Cells were treated as described in (A) and the expression of indicated genes determined by quantitative PCR. All values were normalised to the steady-state, DMSO treated condition (bars 1 and 5). Mean ± S.D. (n = 3) and significance are indicated, unpaired Students t-test: **p ≤ 0.01, ***p ≤ 0.001, n.s. not significant.”

4) These studies were obviously performed in parallel to those described in Jiang et al. JBC 2018. While the authors mention this paper, they should discuss their own results in the context of this other study.

Where appropriate, we have now placed our findings in context of the parallel study (Jiang et al., 2018):

“During preparation of this manuscript, the combined involvement of RNF145 and gp78 in Insig-mediated HMGCR degradation in hamster (CHO) cells was also reported (Jiang et al., 2018), confirming the role for these ligases in other species.”

“In contrast to Jiang et al., who reported a constitutive, sterol-independent association between ectopically expressed RNF145 and Insig-1 or -2, we find that endogenous RNF145 interacts with endogenous Insig-1 in a sterol-dependent manner (Figure 6C), as reported for the interaction of SCAP and HMGCR with the Insig proteins (Lee et al., 2007). The binding of RNF145 to Insig-1 is HMGCR-independent (Figure 6D). Furthermore, in the absence of Insigs, the RNF145-HMGCR association is lost (Figure 6E), implying that the interaction between these two proteins is absolutely Insig-dependent. Therefore, sterols trigger the recruitment of RNF145 to HMGCR via Insigs, leading to HMGCR ubiquitination and degradation. This ability of RNF145 to rapidly bind Insigs following sterol availability supports a key role for this ligase in HMGCR regulation.”

5) Evidence for a RNF145-Insig-HMGCR complex is strong. However, the authors depict RNF145 binding to Insig directly and not HMGCR in Figure 8. Suggest that authors test whether the RNF145-HNGCR interaction requires Insig1/2 to support this model or cite the work of others as support for this specific depiction.

We agree that our model, as depicted in Figure 8, shows RNF145 binding to HMGCR via the intermediary Insig proteins, and we did not exclude a direct interaction between RNF145 and HMGCR. Our initial attempts to assess this issue with endogenous proteins were technically unsuccessful. However, spurred on by the reviewers’ comments, this experiment has now been successful and we now include additional data which supports our model (Figure 6D, E, and Figure 6—figure supplement 1 for validation of the Insig1+2 knockout cell line). Using the immunoprecipitation strategy employed in Figure 6C, we show that (i) RNF145 still binds Insig-1 in the absence of HMGCR (Figure 6D), and importantly, (ii) RNF145 binding to HMGCR is lost upon Insig deletion (Figure 6E).

Figure 6—figure supplement 1 describes how Insig-1+2 knockout cells have been derived and validated. In short, combined CRISPR/Cas9-mediated knockout of both Insigs resulted in constitutive, sterol-independent upregulation of the HMGCR-Clover reporter (see Figure 1F). This strong phenotype was used to enrich for an Insig-1 and -2 depleted population (Figure 6—figure supplement 1A, B(i)) and isolation of ΔInsig-1+2 single cell clones (Figure 6—figure supplement 1B(ii), C).

We reference these findings in the Results section:

“Binding of RNF145 to Insig-1 was HMGCR-independent (Figure 6D), and in the absence of Insigs, RNF145 was unable to bind HMGCR (Figure 6E, see Figure 6—figure supplement 1A-C for generation of Insig-1+2 knockout cells). Insigs are therefore indispensable for the interaction between RNF145 and HMGCR. These findings emphasize the central role of Insig proteins as scaffolds in the sterol-induced engagement of HMGCR by RNF145.”

The legend accompanying Figure 6 has been expanded appropriately:

“(D – E) Insigs mediate binding between RNF145 and HMGCR.

(D) HeLa UBE2G2 knockout cells (ΔUBE2G2 #1) transfected with a pool of 4 sgRNAs targeting HMGCR (gHMGCR) or sgRNA targeting B2M (gCTR), transfected cells enriched by puromycin selection, and sterol-depleted for 20 h, before adding back sterols for 1 h + NMS-873 (10 μM, 1.5 h). Endogenous RNF145 was immunoprecipitated and Insig-1 and HMGCR detected by immunoblot analysis. Non-specific bands are designated by an asterisk (*).

(E) HMGCR-Clover HeLa WT or Insig-1+2 knockout (ΔIn-1+2) cells were transfected with a pool of three gRNAs targeting UBE2G2 (gUBE2G2) and treated as in (D). Non-specific bands are designated by an asterisk (*).”

The generation and validation of the Insig-1+2 double knockout cells is shown in a new supplemental figure (Figure 6—figure supplement 1) with the accompanying figure legend:

“Figure 6—figure supplement 1. Generation of Insig knockout cell lines.

(A) HeLa HMGCR-Clover cells expressing Cas9 were transfected with 4 sgRNAs targeting Insig-1 and Insig-2 (ΔInsig-1+2), and selected with puromycin. Due to the low initial enrichment of Insig-1+2 double knockouts in this experiment, double knockouts were further enriched by two consecutive rounds of FACS (‘sort #1’ and ‘sort #2’), selecting cells with high constitutive HMGCR-Clover expression (‘sorted’, red box) after overnight sterol depletion (SD) and sterol repletion (S) for 2 h. r.u., relative units.

(B) Immunoblotting for endogenous Insig-1 in cells isolated from sort #1 and #2 in (A) (i) and in the Insig-1+2 double knockout clone (ΔIn-1+2 clone) isolated from the enriched population in sort #2 (ii). Insig-2 expression could not be assessed by immunoblotting due to the lack of a working Insig-2 antibody and/or low expression levels in these cells.

(C) Validation of the Insig-1+2 double knockout clone isolated by single-cell sorting from the sort #2 population in (A). Loss of both Insigs was confirmed by presence of sterol-independent, constitutive expression of HMGCR-Clover due to de-repression of the SCAP-SREBP2 complex (red line). HMGCR-Clover expression under steady-state conditions in Insig-1+2 double knockout cells is comparable to that of the sterol-depleted (SD, 20 h) parental WT cell line (blue line). Loss of Insig-1 in the Insig-1/2 double knockout single cell clone was further confirmed by immunoblotting (see (B) and Figure 6E).”

Minor comments:1) Please describe in the results why the p97 inhibitor was included in Figure 6 experiments and whether addition is required for interaction.

We show that RNF145 interacts with HMGCR and Insig proteins in a sterol-dependent manner. Once RNF145 binds HMGCR, it would mediate the ubiquitination and subsequent degradation of its substrate. The VCP inhibitor is therefore necessary to efficiently capture this transient interaction and prevent otherwise rapid HMGCR degradation. A similar strategy has been described to facilitate the efficient enrichment and identification of ERAD substrates (Huang et al., 2018). Of note, the VCP inhibitor can be omitted in UBE2G2 knockout cells since the ubiquitination reaction is blocked by the lack of the E2 enzyme.

2) Figure 8 is a beautiful figure. However, the arrows next to STEROLS in the center are somewhat confusing and appear to indicate that sterols are moving between the top and bottom panels. Suggest writing "Low sterol" and "High sterol" or something similar to improve figure clarity.

Thank you for this suggestion; we have now clarified the labelling in Figure 8.

3) To improve clarity, please describe in more detail the biology behind the US2-MHC1 assay used to validate TRC8 knockout.

We are grateful for pointing out this omission and have now included a more detailed description of the results in the legend to Figure 7B—figure supplement 2:

“(B) TRC8 knockdown was confirmed by US2-mediated TRC8-dependent downregulation of MHC-I. HeLa cells transiently expressing either B2M sgRNA (gCTR) or TRC8-sepcific sgRNAs (gTRC8) were selected for puromycin resistance and transduced with a lentiviral US2 and/or TRC8 construct 5 days *post* transfection. Cell-surface MHC-I staining and FACS analysis were performed on day 10 *post* transfection. US2 is a herpes cytomegalovirus (HCMV)-encoded gene product which directs TRC8 to degrade MHC-I in the ER lumen (Stagg et al., 2009; Hsu et al., 2015). Loss of TRC8 renders cells resistant to US2-mediated MHC-I loss (green line in histogram) and therefore serves as a functional readout for TRC8 status.”

4) Introduction paragraph three is missing a word, "despite confirming a for gp78"

Thank you, the amended sentence now reads:

“However, these findings remain controversial as, despite confirming a role for gp78 in the regulation of Insig-1 (Lee et al., 2006; Tsai et al., 2012), an independent study found no evidence for either gp78 or TRC8 in the sterol-induced degradation of HMGCR (Tsai et al., 2012). Therefore, the E3 ligase(s) responsible for the sterol-accelerated degradation of HMGCR remain disputed.”

5) Results first paragraph subheading, typo HMMGCR.

Thank you, we have corrected this typographical error to “HMGCR”.